# Characteristics of methane emissions from alpine thermokarst lakes on the Tibetan Plateau

Guibiao Yang[1], Zhihu Zheng[1,2], Benjamin W. Abbott [3], David Olefeldt [4], Christian Knoblauch [5], Yutong Song[1,2], Luyao Kang[1,2], Shuqi Qin[1], Yunfeng Peng [1] & Yuanhe Yang [1,2] ✉

Understanding methane ($CH_4$) emission from thermokarst lakes is crucial for predicting the impacts of abrupt thaw on the permafrost carbon-climate feedback. However, observational evidence, especially from high-altitude permafrost regions, is still scarce. Here, by combining field surveys, radio- and stable-carbon isotopic analyses, and metagenomic sequencing, we present multiple characteristics of $CH_4$ emissions from 120 thermokarst lakes in 30 clusters along a 1100 km transect on the Tibetan Plateau. We find that thermokarst lakes have high $CH_4$ emissions during the ice-free period ($13.4 \pm 1.5$ mmol m$^{-2}$ d$^{-1}$; mean $\pm$ standard error) across this alpine permafrost region. Ebullition constitutes 84% of $CH_4$ emissions, which are fueled primarily by young carbon decomposition through the hydrogenotrophic pathway. The relative abundances of methanogenic genes correspond to the observed $CH_4$ fluxes. Overall, multiple parameters obtained in this study provide benchmarks for better predicting the strength of permafrost carbon-climate feedback in high-altitude permafrost regions.

Soils in high-latitude and high-altitude permafrost regions store more than 50% of world's soil organic carbon (SOC) in about 15% of the global land area[1,2] and play an important role in the global carbon cycle. These permafrost regions are experiencing higher rates of climate warming compared to other parts of the world[3]. This rapid warming is triggering widespread gradual and abrupt permafrost thaw[4] and subsequent release of carbon to the atmosphere in the form of carbon dioxide ($CO_2$) and methane ($CH_4$), potentially acting as a strong positive feedback to climate warming[5–7]. Thermokarst lake formation, a common abrupt thaw process, occurs due to the melting of excess ground ice in areas of permafrost degradation[8]. Such lakes are the most widespread feature due to abrupt permafrost thaw, and cover 7% of permafrost regions[9]. Due to their anaerobic environment, thermokarst lakes can be hot spots for $CH_4$ emissions[10–12], but in-situ measurements are sparse, especially

from high-altitude permafrost regions, hampering our ability to assess the impact of abrupt thaw on the permafrost carbon cycle[10,12,13]. Therefore, improved understanding of $CH_4$ emissions from alpine thermokarst lakes is crucial for predicting permafrost carbon-climate feedback in this climate-sensitive region.

The Tibetan Plateau is the largest alpine permafrost region in the world (Fig. 1a), accounting for approximately 75% of the total alpine permafrost area in the Northern Hemisphere[14]. Similar to high-latitude permafrost regions, this region has experienced fast climate warming and extensive permafrost thaw[3,15], which has triggered the widespread expansion of thermokarst lakes (Fig. 1b, c) and other types of abrupt permafrost thaw[16]. The number of thermokarst lakes in this permafrost region is estimated to be 161,300, with a total area of ~2800 km$^2$ [17]. Most of the lakes (~80%) are located in alpine grasslands which can be

[1]State Key Laboratory of Vegetation and Environmental Change, Institute of Botany, Chinese Academy of Sciences, Beijing 100093, China. [2]University of Chinese Academy of Sciences, Beijing 100049, China. [3]Department of Plant and Wildlife Sciences, Brigham Young University, Provo, UT 84602, USA. [4]Department of Renewable Resources, University of Alberta, Edmonton, Alberta T6G 2H1, Canada. [5]Institute of Soil Science, University of Hamburg, 20146 Hamburg, Germany. ✉e-mail: yhyang@ibcas.ac.cn

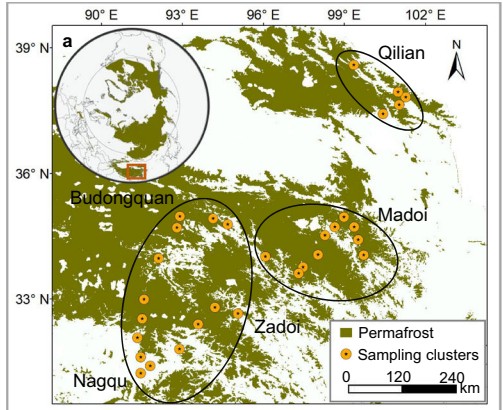

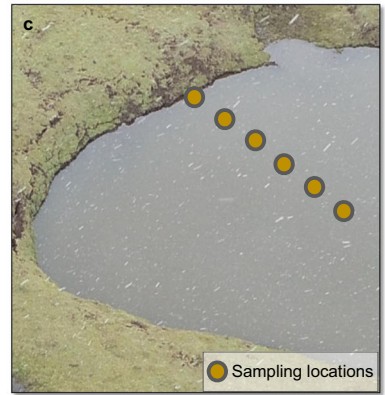

**Step 1** **Selecting sampling clusters:** 30 clusters were located in three representative permafrost regions (Madoi, Budongquan-Nagqu-Zadoi and Qilian sections)

**Step 2** **Sampling at each cluster:** Four thermokarst lakes were selected to cover different lake sizes at each cluster. In each lake, 4 to 6 sampling locations were distributed from the shore to the center.

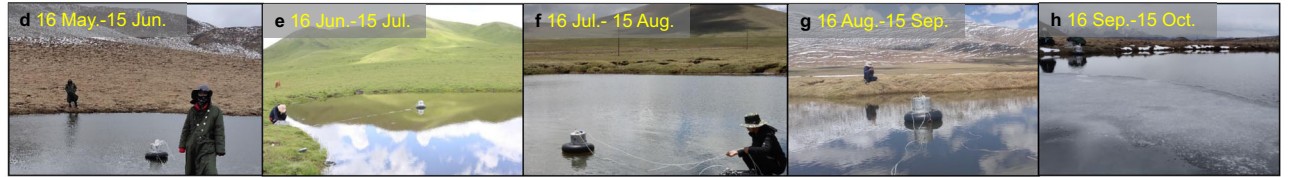

**Step 3** **Repeated sampling at each cluster during the ice-free period:** Each lake was sampled five times at monthly intervals during the ice-free period from mid-May to mid-October of 2021 to explore seasonal variations of $CH_4$ and $CO_2$ fluxes.

**Fig. 1 | The flow chart of the sampling campaign.** Our field sampling consists of the following three key steps. First, we choose 30 clusters of thermokarst lakes along a 1100 km transect on the Tibetan Plateau (**a**). Second, multiple locations within multiple lakes are selected at each cluster to eliminate spatial heterogeneity. In particular, four thermokarst lakes with different sizes are selected at each cluster (**b**). In each lake, 4 to 6 sampling locations are distributed from the shore to center (**c**), and at each location flux measurements are taken and averaged to estimate the $CH_4$ and $CO_2$ flux from the lake. Finally, each lake is sampled five times at monthly intervals during the ice-free period from mid-May to mid-October of 2021 to explore seasonal variation of $CH_4$ or $CO_2$ fluxes (**d**–**h**). In-situ $CH_4$ and $CO_2$ fluxes are determined using an opaque lightweight floating chamber equipped with a closed loop to a near-infrared laser $CH_4/CO_2$ analyzer (GLA231-GGA, ABB., Canada). In (**a**), the permafrost map of the Northern Hemisphere is obtained from the National Snow & Ice Data Center[85]. Spatial distribution of permafrost on the Tibetan Plateau is derived from Zou et al.[86]. The ellipses indicate the three representative permafrost regions in our study area, including the Madoi, Budongquan-Nagqu-Zadoi and Qilian sections. Photos are taken by G.Y.

subdivided into alpine steppe, alpine meadow and swamp meadow[17]. Given that this permafrost region stores a large amount of SOC (15.3–46.2 Pg carbon in the top 3 m of soils; 1 Pg = $10^{15}$ g)[18–20], permafrost thaw could facilitate the rapid microbial decomposition of organic matter, leading to substantial carbon emissions[20,21]. Additionally, the unique environmental conditions, characterized by lower air pressure and oxygen concentration due to the high elevation[15,22], could be beneficial for $CH_4$ production. Thermokarst lakes in this permafrost region are thus expected to behave as hot spots for $CH_4$ emissions[23,24]. However, compared with the Arctic permafrost region, our understanding of $CH_4$ emissions from alpine thermokarst lakes is limited. Specifically, relatively little is known about $CH_4$ fluxes, the contribution of old carbon from thawing permafrost, methanogenic pathways ($CO_2$ reduction versus acetate fermentation), and microbial characteristics (methanogenic functional genes and communities). These knowledge gaps prevent accurate prediction of the magnitude of carbon-climate feedback in this alpine permafrost region.

In this context, we conducted a large-scale sampling campaign across 120 thermokarst lakes in 30 clusters (four lakes in each cluster: 4 lakes/cluster × 30 clusters) along a 1100 km transect on the Tibetan Plateau (Fig. 1a, b). Each lake was sampled five times at monthly intervals during the ice-free period from mid-May to mid-October of 2021 (Fig. 1d–h), with each campaign lasting ~25 days. We measured $CH_4$ fluxes to the atmosphere using a portable opaque dynamic chamber (Supplementary Fig. 1). Our results show that the thermokarst lakes on the Tibetan Plateau are an important $CH_4$ source (regional $CH_4$ emission: 76.6 Gg ($10^9$ g) $CH_4$ yr$^{-1}$ with a mean flux of 13.4 ± 1.5 mmol m$^{-2}$ d$^{-1}$ during the ice-free period; hereafter, values are reported as mean ±

standard error (SE) unless stated otherwise). Ebullition is the main pathway for $CH_4$ release, contributing to 84% of $CH_4$ emissions. Radio- and stable-carbon analyses show that old carbon is not the dominant source for $CH_4$ production, with $CH_4$ being derived mainly from the hydrogenotrophic pathway in most of the sampled lakes. The relative abundances of methanogenic genes correspond to the in-situ $CH_4$ fluxes. These findings lay the groundwork for a comprehensive understanding of $CH_4$ emissions in high-altitude thermokarst lakes.

## Results and discussion
### $CH_4$ emissions across sampling clusters
Across the 30 sampled clusters, $CH_4$ concentrations ranged from 107.1 to 159.4 nmol L$^{-1}$ with a mean of 136 ± 30 nmol L$^{-1}$ ($n$ = 30; Supplementary Fig. 2a). $CH_4$ was supersaturated relative to the local atmosphere in all the studied thermokarst lakes across the clusters with a mean value of 2921 ± 62% (ranging from 2393 to 3719%; $n$ = 30). $CH_4$ fluxes had high spatial variability within the 30 clusters, with values ranging from 0.1 to 39.2 mmol m$^{-2}$ d$^{-1}$ (Fig. 2). The lowest values occurred in thermokarst lakes located in alpine steppe (8.7 ± 3.0 mmol m$^{-2}$ d$^{-1}$) and the highest in alpine meadow (16.1 ± 1.7 mmol m$^{-2}$ d$^{-1}$). The $CH_4$ fluxes also displayed temporal variability during the sampling period (Supplementary Fig. 3a). The maximum monthly mean $CH_4$ flux was observed during the period of mid-June to mid-July (30.5 ± 4.9 mmol m$^{-2}$ d$^{-1}$), while the minimum occurred at the end of ice-free season (7.6 ± 1.5 mmol m$^{-2}$ d$^{-1}$), possibly due to the low temperature.

The mean $CH_4$ flux during the ice-free season was 13.4 ± 1.5 mmol m$^{-2}$ d$^{-1}$. This value is at the high end of the range reported from Arctic thermokarst water bodies regarded as hot spots for $CH_4$

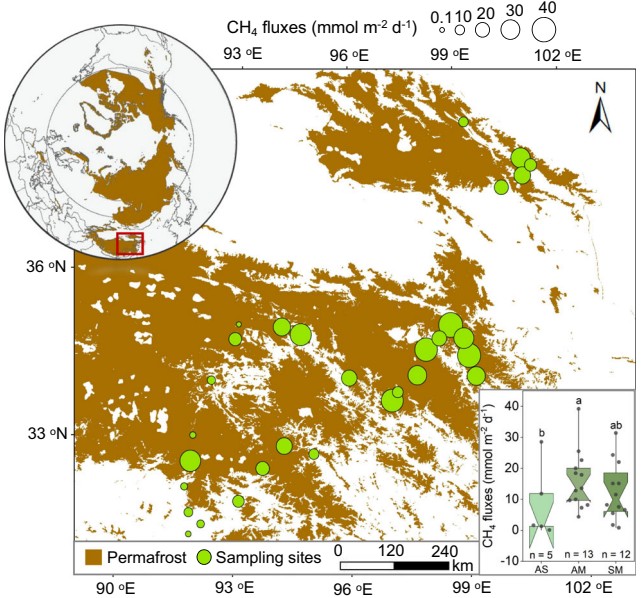

**Fig. 2 | CH$_4$ fluxes in alpine thermokarst lake on the Tibetan Plateau.** Bubble size is proportional to the value of the CH$_4$ flux at each cluster, with a larger size representing a higher value. The background permafrost maps of the Northern Hemisphere and the Tibetan Plateau are derived from the National Snow & Ice Data Center[85] and Zou et al.[86], respectively. The inset shows the comparison of CH$_4$ fluxes in thermokarst lakes located in the three grassland types. AS, AM and SM represent alpine steppe, alpine meadow and swamp meadow, respectively. Box plots present the 25th and the 75th quartile (interquartile range), and whiskers indicate the data range among thermokarst lakes located in AS ($n = 5$), AM ($n = 13$) and SM ($n = 12$), respectively. The notches are the medians with 95% confidence intervals. Observed values are shown as black dots. Significant differences are denoted by different letters (one-way ANOVAs with two-sided Tukey's HSD multiple comparisons, $p = 0.049$).

release[13,25]. During our field sampling campaign, we also measured CO$_2$ emissions (see Supplementary Note 1 for detailed information). Our results showed that the contribution of CH$_4$ to total carbon emissions (CH$_4$ + CO$_2$) from thermokarst lakes was 7.3% (Fig. 3a). If considered in terms of CO$_2$-equivalent (CO$_2$-e) emissions, mean CH$_4$ flux during the ice-free season was estimated to be 136.8 CO$_2$-e mmol m$^{-2}$ d$^{-1}$ (a 28-fold higher global warming potential relative to CO$_2$ over 100 years)[26], which was of the same order of magnitude as the CO$_2$ emissions (170.4 mmol CO$_2$ m$^{-2}$ d$^{-1}$; Supplementary Fig. 4) and approximately 44.6% of total CO$_2$-e emissions (Fig. 3b). In conjunction with the total area of thermokarst lakes (~2800 km$^2$)[17], these findings demonstrate that alpine thermokarst lakes are hot spots of CH$_4$ emission, and also highlight the importance of CH$_4$ fluxes in the total carbon emissions from alpine thermokarst lakes. The high CH$_4$ fluxes can be attributed to two characteristics of the Tibetan Plateau. First, atmospheric oxygen concentration across our study area is low due to the high elevation (between 3279 and 5014 m above sea level at our study sites), which can cause low dissolved oxygen concentration in all water bodies, including thermokarst lakes (4.3 ± 0.2 mg L$^{-1}$; $n = 30$; Supplementary Table 1). Low dissolved oxygen concentration can stimulate CH$_4$ production, and thus increase the CH$_4$ flux[27]. Second, the thermokarst lakes across our study area are mostly shallow (depth range: ~0.2-3.7 m; Supplementary Table 1)[22]. The shallow water column allows more rapid exchange with the atmosphere and less time for CH$_4$ removal by microbial oxidation[28], and is thus conducive to the release of CH$_4$.

## CH$_4$ diffusion and ebullition fluxes
To isolate the main pathways of CH$_4$ release, we quantified both diffusion and ebullition fluxes. Ebullition occurred at all sampling clusters

and exhibited a large spatiotemporal variability (Fig. 4a). Throughout the ice-free season, CH$_4$ ebullition fluxes were highest from mid-June to mid-July with a mean value of 29.7 ± 4.7 mmol m$^{-2}$ d$^{-1}$ ($n = 30$; Supplementary Fig. 3a). During the ice-free period, mean CH$_4$ diffusion across all the sampled clusters varied from 0.1 to 7.9 mmol m$^{-2}$ d$^{-1}$ while ebullition varied from 0.03 to 31.4 mmol m$^{-2}$ d$^{-1}$. The mean values were 2.1 ± 0.3 mmol m$^{-2}$ d$^{-1}$ for diffusion and 11.2 ± 1.5 mmol m$^{-2}$ d$^{-1}$ for ebullition ($n = 30$; Supplementary Fig. 3a). The maximum ebullition flux (200.8 mmol m$^{-2}$ d$^{-1}$) was recorded at the shore of one of the thermokarst lakes, where micro-bubbles were visible (Supplementary Movie 1). The contribution of ebullition to lake CH$_4$ fluxes showed no significant difference among these three grassland types in which the thermokarst lakes are mainly distributed ($p = 0.07$; Supplementary Fig. 5a). Overall, alpine thermokarst lakes on the Tibetan Plateau were of high ebullition fluxes which constituted ~84% of the total CH$_4$ fluxes (diffusion plus ebullition fluxes; Fig. 4a).

The high contribution of ebullition to the total CH$_4$ flux might be potentially explained by low atmospheric pressure on the Tibetan Plateau. Due to the high elevation, atmospheric pressure across the study area has a mean value of 60.9 kPa: roughly three-fifths of that at sea level (Supplementary Table 1). Two ways in which this low atmospheric pressure could result in a larger contribution of ebullition to the total CH$_4$ flux. On the one hand, according to the Henry's law, the lower atmospheric pressure causes lower CH$_4$ solubility in the water[29], which can be unfavorable for CH$_4$ diffusion in the water column and thus impels CH$_4$ to be transported from the lakebed to the atmosphere in the form of bubbles. On the other hand, atmospheric pressure can directly affect bubble formation. Bubbles containing CH$_4$ occupy vertical tubes within the lake sediments[30]. The lower air pressure will benefit the vertical expansion of these bubbles and promote their escape from the lake sediment[31], and is thus associated with greater ebullition of CH$_4$[32]. Consistently, the proportion of ebullition to total CH$_4$ fluxes was negatively associated with air pressure ($R^2 = 0.42$, $p < 0.001$; Supplementary Fig. 6a) but positively correlated with elevation across sampling clusters ($R^2 = 0.42$, $p < 0.001$; Supplementary Fig. 6b). In addition, the shallowness of alpine thermokarst lakes on the Tibetan Plateau (Supplementary Table 1) could contribute to the high relative ebullition because shallow water creates less hydrostatic pressure[22], increasing the formation and release of gas bubbles from lake sediments[31,33].

## Radiocarbon age of CH$_4$ emissions
To estimate the contribution of old carbon to CH$_4$ emissions, we collected ebullition gas samples with floating plastic bubble traps and determined the CH$_4$ radiocarbon age. The results showed relatively young CH$_4$ radiocarbon age, ranging from −360 to 3810 years before present (yrs BP; Fig. 4b). The mean CH$_4$ radiocarbon age was only 325.8 yrs BP, and 46% of thermokarst lakes had modern (defined here as created after 1950) age of CH$_4$ emissions (Supplementary Table 2). Only two lakes had higher CH$_4$ radiocarbon ages than 1000 yrs BP, while radiocarbon ages were between modern and 1000 yrs BP in the remaining 22 samples. These observations indicated that old carbon was not the dominant source for CH$_4$ production in most thermokarst lakes on the Tibetan Plateau. Potential explanations for the relatively low contribution of old carbon to CH$_4$ fluxes observed in this study could be associated with the young permafrost carbon in this study region. It has been reported that permafrost on the Tibetan Plateau forms relatively recently compared to other permafrost regions[34,35], which may lead to the situation that the frozen carbon is also relatively young. This deduction is supported by the measured average radiocarbon age of surface permafrost below the active layer at 24 sites across the Tibetan Plateau (6100 ± 880 yrs BP; $n = 24$; Supplementary Fig. 7; see Supplementary Note 2 for details of radiocarbon age measurements). Therefore, the relatively low contribution of old carbon to CH$_4$ fluxes in alpine thermokarst lakes could be attributed to the young

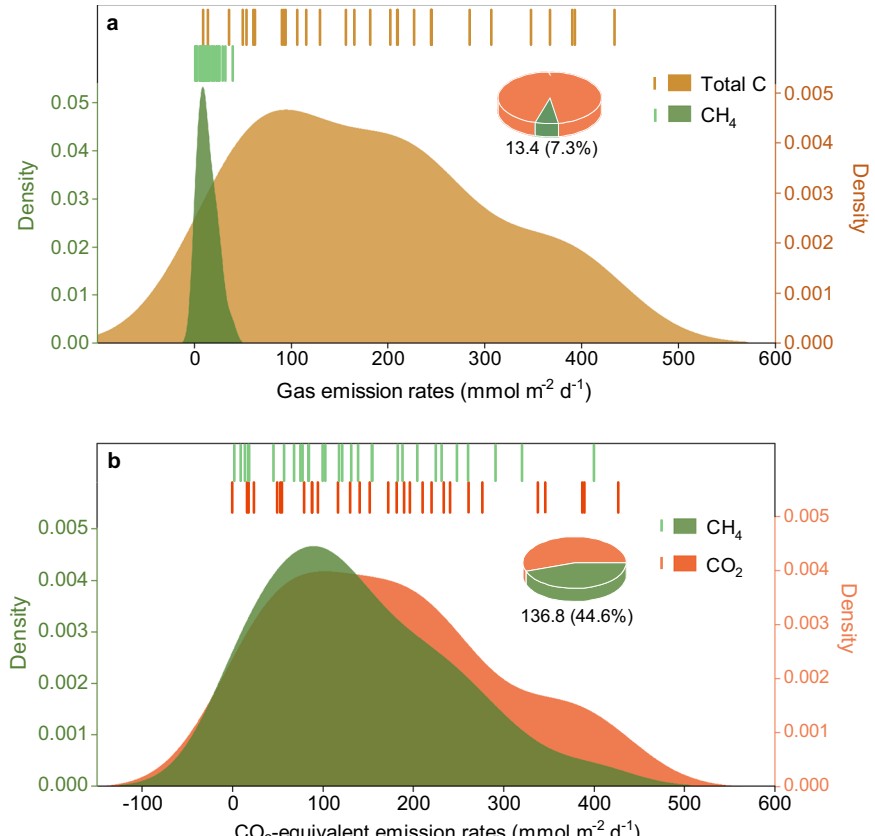

**Fig. 3 | Contribution of CH₄ to total carbon emissions from the investigated thermokarst lakes.** Panels (**a**, **b**) represent the density of carbon fluxes and CO₂-equivalent emmissions, respectively. The lines indicate the fluxes from four thermokarst lakes at each cluster during the measurement period. The pie charts show the contribution of CH₄ fluxes to total carbon emissions and CO₂-equivalent emissions. Mean CH₄ flux during the ice-free season and its CO₂-equivalent emissions are shown outside parentheses, and the corresponding contribution to total carbon emissions and CO₂-equivalent emissions are presented in parentheses, respectively.

permafrost carbon in this study region. In addition, thermokarst lakes on the Tibetan Plateau are characterized by the small surface area[22], and small lakes have a high perimeter to surface area ratio, which potentially increase terrestrial loading of organic matter from surrounding plants and surface soils. This terrestrial organic matter is dominated by modern carbon[36], and can thus stimulate modern carbon emissions from thermokarst lakes[37]. The last but not the least, some of the studied lakes may have developed following thermokarst processes a long time ago or even not be thermokarst lakes. This might also be part of the explanation of the modern age of emitted CH₄.

## CH₄ production pathway

To estimate the relative contribution of two major pathways of methanogenesis (CO₂ reduction and acetate fermentation), we measured the $\delta^{13}C$ of CH₄ and CO₂ in bubble samples to calculate the apparent carbon fractionation factor ($\alpha_C$); an indicator of the CH₄ production pathway (see "Methods"; $\alpha_C > 1.055$ indicates CO₂ reduction; $\alpha_C < 1.04$ indicates acetate fermentation)[38]. The $\delta^{13}C$-CH₄ had a mean value of $-72.5 \pm 1.1$‰ ($n = 29$), and the $\delta^{13}C$-CO₂ ranged between $-0.5$‰ and $-22.9$‰ with a mean of $-13.4$‰ (Supplementary Table 2). Gas samples exhibited high $\alpha_C$ values, ranging from 1.052 to 1.079 ($1.064 \pm 0.001$, $n = 29$). Only four samples had $\alpha_C$ values between 1.04 and 1.055, while the rest had $\alpha_C$ values higher than 1.055 (Fig. 4c), indicating that CO₂ reduction dominated CH₄ production in alpine thermokarst lakes on the Tibetan Plateau. Furthermore, the $\alpha_C$ values showed no significant difference among the three grassland types in which thermokast lakes are mainly distributed ($p = 0.52$; Supplementary Fig. 5b).

High $\alpha_C$ values may be attributed to the alkaline and saline conditions across our study area (Supplementary Table 1). Specifically, high pH could stimulate the dissociation of acetic acid into its anion form (CH₃COO⁻), which could then inhibit transmembrane diffusion and prevent the transportation of acetate[39]. Therefore, despite the accumulation of acetic acid, acetoclastic methanogenesis is likely to be less energetically favorable than hydrogenotrophic methanogenesis under alkaline conditions, leading to the high contribution of the CO₂ reduction pathway to CH₄ production[40]. Beside the alkaline conditions, saline environment-associated methanogenic substrates may be another potential driver for high $\alpha_C$ values. In saline environment, methanol has been reported to be a methanogenic precursor and can serve as substrates for CH₄ production[41]. Moreover, methanol-derived methanogenesis is usually accompanied by the highly depleted $\delta^{13}C$-CH₄ values[42]. Consequently, methanol-dependent methanogenesis may also be responsible for the high $\alpha_C$ values from alpine thermokarst lakes on the Tibetan Plateau due to their saline environment (ranging from 0.2 to 2.6 ppt; Supplementary Table 1).

## Methanogenic microorganisms

To evaluate the potential differences in methanogenic microorganisms from thermokarst lakes in the three ecosystem types, we analyzed methanogenic functional genes and communities in surface sediment samples (0–15 cm) using metagenomic sequencing. Multiple functional genes of methanogens were more abundant in the thermokarst lakes located in alpine meadow and swamp meadow (Supplementary Fig. 8). This result suggested that there were higher potentials for CH₄ production in thermokarst lakes located in these two grassland types,

which matched with higher $CH_4$ fluxes observed in the field at these locations (Fig. 2). The largest differences in relative abundance were observed for the *mcr* gene, with mean value being 4.1-fold and 3.2-fold higher in thermokarst lakes distributed in alpine meadow and swamp meadow than those located in alpine steppe, respectively

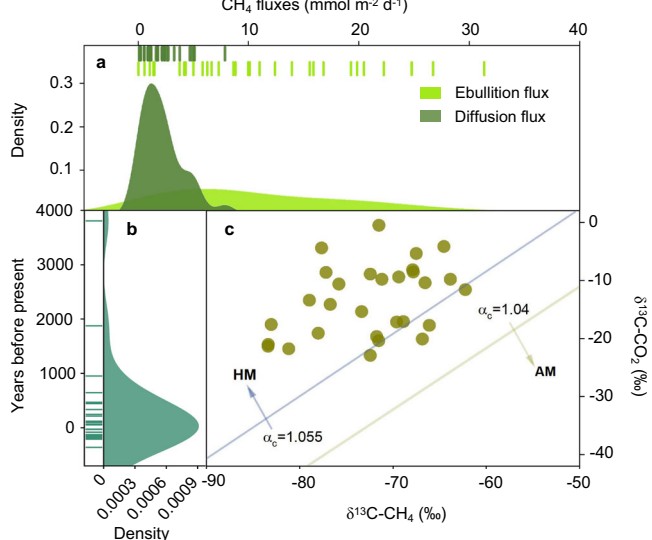

**Fig. 4 | CH₄ ebullition and diffusion fluxes, radiocarbon age and production pathway in alpine thermokarst lakes on the Tibetan Plateau.** In (**a**), the corresponding values of light green and dark green lines represent the ebullition and diffusion $CH_4$ fluxes across 30 clusters, respectively. Line shows the mean $CH_4$ flux for the four thermokarst lakes at each cluster. In (**b**), line indicates the radiocarbon age of surface permafrost below the active layer at each sampling site (*n* = 24). In (**c**), the light blue and light orange lines are the apparent carbon fractionation factor ($\alpha_C$) values of 1.04 and 1.055. The $\alpha_C$ values indicate the pathway of $CH_4$ production, with $\alpha_C > 1.055$ suggesting that $CH_4$ is mainly produced by $CO_2$ reduction (hydrogenotrophic methanogenesis, HM), and $\alpha_C < 1.055$ suggesting that $CH_4$ is produced increasingly by acetate fermentation (acetoclastic methanogenesis, AM). $^{14}C$ and $\delta^{13}C$ isotopic signatures were measured in the bubble gas from only 24 and 29 lakes respectively, due to the limited volume of gas samples that could be collected in the field.

(Supplementary Fig. 8). This result was confirmed by the validated *mcrA* gene predicted by contigs with length ≥ 1000 bp (Fig. 5a).

With regard to the sediment methanogenic community composition of the thermokarst lakes, there were five dominant methanogenic classes: *Methanomicrobia*, *Methanobacteria*, *Thermoplasmata*, *Methanococci*, and *Methanopyri* (Fig. 5b). The genus composition diagrams showed that *Methanosarcina* within *Methanosarcinaceae* order was the most abundant methanogenic genus, accounting for about 65% of all methanogens. It has been reported that *Methanosarcina* has a high volume-to-surface ratio with a large cell size and spherical form. Together with the formation of clusters, this leads to low levels of diffusion per unit cell mass[43]. Accordingly, *Methanosarcina* is relatively tolerant of adverse environmental conditions compared to other methanogens[44]. Moreover, the *Methanosarcina* genus contains cytochromes and methanophenazine (a functional menaquinone analog), which enable methanogens to conserve energy via membrane-bound electron transport chains so as to maintain high growth yields[45]. Overall, *Methanosarcina* can be competitive at low temperature, alkaline, and saline conditions (Supplementary Table 1), potentially contributing to high $CH_4$ emission rates from alpine thermokarst lakes on the Tibetan Plateau.

## Regional estimates

To upscale our lake-level measurements to regional efflux estimates, we conducted a Monte Carlo analysis to randomly sample thermokarst lake $CH_4$ flux for each grassland type from a normal distribution around the mean. We then weighted the $CH_4$ flux by the corresponding area of thermokarst lakes in each grassland type to calculate the regional flux. Total $CH_4$ emissions were estimated to be 76.6 Gg $CH_4$ $yr^{-1}$. The contribution of $CH_4$ to total carbon emissions was approximately 9.2% (Table 1), which was 1.4–8.4-fold greater than the mean contribution from lakes globally[28,38–40]. The 100-year global warming potential of $CH_4$ was 2144.7 Gg $CO_2$-e $yr^{-1}$, which was approximately equivalent to lake $CO_2$ emissions estimated in the same way (2084.7 Gg $CO_2$ $yr^{-1}$; Table 1). $CH_4$ emissions from thermokarst lakes are often ignored when evaluating the regional $CH_4$ balance across Tibetan alpine grasslands[17]. However, our results illustrated that $CH_4$ emissions from thermokarst lakes could offset 15.3% of the $CH_4$ uptake from alpine steppe and meadow which cover -95% of alpine grasslands on

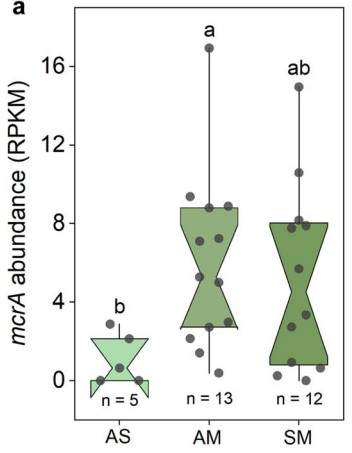

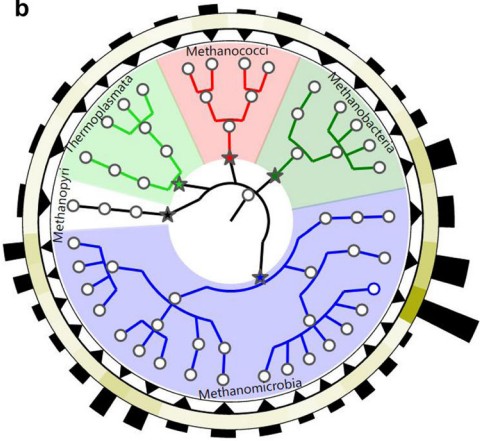

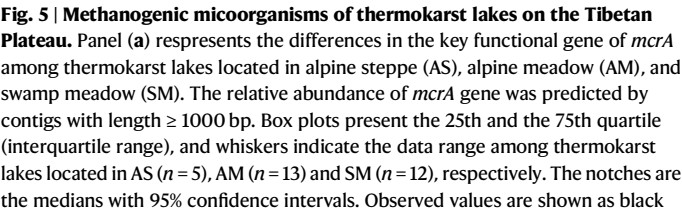

**Fig. 5 | Methanogenic micoorganisms of thermokarst lakes on the Tibetan Plateau.** Panel (**a**) respresents the differences in the key functional gene of *mcrA* among thermokarst lakes located in alpine steppe (AS), alpine meadow (AM), and swamp meadow (SM). The relative abundance of *mcrA* gene was predicted by contigs with length ≥ 1000 bp. Box plots present the 25th and the 75th quartile (interquartile range), and whiskers indicate the data range among thermokarst lakes located in AS (*n* = 5), AM (*n* = 13) and SM (*n* = 12), respectively. The notches are the medians with 95% confidence intervals. Observed values are shown as black

dots. Significant differences are denoted by different letters (one-way ANOVAs with two-sided Tukey's HSD multiple comparisons, *p* = 0.038). Panel (**b**) shows methanogenic taxonomic infromation. The colors in the inner ring represent the different taxa. The triangles in the first ring indicate relative abundance with ≥ 5 per 10,000 (upper triangle) or <5 per 10,000 (lower triangle). The mean relative abundances for all samples are shown in the second ring and pillars, where color depth and height are proportional to the cubic root of relative abundance.

**Table 1 | Regional carbon and $CO_2$-equivalent emissions ($CH_4$ and $CO_2$) from alpine thermokarst lakes on the Tibetan Plateau**

| Target variables | $CH_4$ emissions | $CO_2$ emissions |
|---|---|---|
| Carbon emissions (Gg C yr$^{-1}$) | 57.4 (0.8–114.1) | 568.6 (17.5–1119.7) |
| $CO_2$-equivalent emissions (Gg $CO_2$-e yr$^{-1}$) | 2144.7 (28.9–4260.5) | 2084.7 (205.8–4105.5) |

5th–95th percentiles are presented in parentheses.

the Tibetan Plateau[46,47]. Further, thermokarst lake $CH_4$ emissions were equivalent to ~50–150% of the $CH_4$ emissions from swamp meadow which occupies ~5% of Tibetan alpine grasslands[18,47]. The incorporation of $CO_2$ fluxes gives an estimate of the overall carbon emissions ($CH_4 + CO_2$) in thermokarst lakes of 4.2 Tg ($10^{12}$ g) $CO_2$-e yr$^{-1}$ (Table 1), which could offset 11.9% of net carbon sink in Tibetan alpine grasslands (37.1 Tg $CO_2$-e yr$^{-1}$)[48]. Taken together, these results demonstrate that any assessment of the carbon budget in this climate-sensitive region is incomplete without considering the significant carbon emissions from thermokarst lakes.

Although this study advances our understanding of $CH_4$ release from thermokarst lakes on the Tibetan Plateau, it does have some limitations. First, carbon emissions during the ice-cover period were not considered in this study. To make a rough evaluation of ice-cover carbon emission, we applied the average percent contribution of ice-cover to annual carbon emissions from high-latitude lakes ($CO_2$: 17%; $CH_4$: 27%)[49]. These increased annual emissions estimate from alpine thermokarst lakes on the Tibetan Plateau to 427.0 Gg $CO_2$ yr$^{-1}$ and 28.3 Gg $CH_4$ yr$^{-1}$, respectively. Nevertheless, given the potential differences between high-latitude and high-altitude permafrost regions, such as climate, thermokarst lake depth and ice-cover duration[34], the annual contribution of carbon emissions during the ice-cover period observed from high-latitude lakes may not be simply applied to thermokarst lakes in high-altitude permafrost region. More measurements during the ice-cover period are thus needed to further elucidate the role of thermokarst lakes in the regional carbon budget. Second, uncertainties exist in the area of the thermokarst lakes used in the regional carbon budget. Particularly, newly formed thermokarst lakes and thermokarst lakes located in desert and barren land were not considered, but they do have potential to contribute to $CH_4$ emissions[50]. Meanwhile, theomarkast lake map used in this study suffers from uncertainties because it is based on low resolution and single satellite images and without consideration of ground-ice content[17]. It is thus essential to incorporate multi-satellite, higher spatial resolution remote sensing data (e.g. GF-2 and Planetscope) and ground-ice content to re-evaluate thermokarst lake area and its temporal dynamics in the future. Based on the updated lake distribution and the expansion rate, an improved estimation by considering thermokarst lakes distributed in other ecosystems and fresh thermokarst lakes is needed to obtain more accurate prediction of carbon release from alpine thermokarst lakes. Third, the development of thermokarst lake is not taken into account in this study. The studied lakes may cover the various thermokarst development stages or even non-thermokarst lakes, which can result in the uncertainty of the regional carbon emissions. Therefore, additional attention should be paid for the development of thermokarst lakes to further advance our understanding of $CH_4$ emissions in alpine thermokarst lakes on the Tibetan Plateau.

In summary, this study offers an insight into the spatial patterns, sources and microbial characteristics of $CH_4$ emissions from alpine thermokarst lakes on the Tibetan Plateau. Our results indicated that thermokarst lakes were an important but under-quantified $CH_4$ source in the Tibetan alpine permafrost region. Considering that the expansion of thermokarst lake landscapes will accelerate under future climate warming, their potential contribution to the regional $CH_4$ budget

may increase substantially. Moreover, this study evaluated the landscape-level radiocarbon ages of thermokarst lake $CH_4$ emissions in this alpine permafrost region. Our results illustrated that old carbon was not the dominant source for $CH_4$ production from alpine thermokarst lakes on the Tibetan Plateau. This finding is in contrast to studies from the high-latitude permafrost region where old carbon contributes significantly to $CH_4$ emissions upon permafrost thaw[51]. Hence, when modeling permafrost carbon-climate feedback across our study area, more efforts should be put into accurately predicting the effect of permafrost thaw on the spatial extent of thermokarst lakes, rather than on the rate of anaerobic decomposition of previously frozen soil. Furthermore, our results demonstrated that methanogenic functional genes corresponded with the in-situ $CH_4$ fluxes, suggesting that methanogenesis might be the potential driver of $CH_4$ emissions from alpine thermokarst lakes. Overall, multiple parameters observed in this study can function as benchmarks for better predicting permafrost carbon-climate feedback.

## Methods
### Study area
The Tibetan Plateau, the largest alpine permafrost region in the world, has a mean elevation of over 4000 m above sea level. Discontinuous and sporadic permafrost are widely distributed across the region[52]. Approximately $1.06 \times 10^6$ km$^2$ of this region is underlain by permafrost, accounting for 40% of the overall plateau[14]. The permafrost mainly formed in the late Pleistocene during the Last Glaciation Maximum and the Neoglaciation period[35]. The current active layer thickness across the plateau is ~1.9 m, but this is deepening at a rate of ~1.3 cm yr$^{-1}$ based on observations from 1981 to 2010[53,54]. The climate is characterized as cold and dry[14]. Mean annual temperature ranges from −2.9 to 7.0 °C and mean annual precipitation varies from 129 to 590 mm, with approximately 90% of the precipitation falling during the growing season (late May to early October). The arid and semiarid climate has suppressed the development of ground-ice, leading to a relatively low ice content with a percentage by weight of 12.2% in this permafrost region[34,55]. The dominant vegetation types include alpine steppe, alpine meadow, and swamp meadow, with the corresponding dominant species being *Stipa purpurea* and *Carex moorcroftii*, *Kobresia pygmaea* and *K. humilis*, and *K. tibetica*, respectively. The plateau has experienced rapid climate warming, causing the formation and expansion of thermokarst lakes in various vegetation types (Supplementary Fig. 2c–e)[56]. The distribution of thermokarst lakes is dominated by lakes with small surface area (<10,000 m$^2$) and shallow water which account for ~80% of the total number[17]. The mean ice-free duration of lakes across the study area is around 200 days[57].

### Flux and environmental measurements
In this study, we selected 30 clusters of thermokarst lakes for carbon flux measurements along a 1100 km transect on the Tibetan Plateau. The 30 clusters were evenly located in three representative permafrost regions (10 sites in the Madoi section on the eastern plateau, 15 sites in the Budongquan-Nagqu-Zadoi section, and 5 sites in the Qilian section on the northeastern plateau in the central part of the plateau; Fig. 1a). At each cluster, four thermokarst lakes were selected to cover different lake sizes. A total of 120 thermokarst lakes were thus sampled (30 clusters × 4 lakes/cluster). In each lake, 4–6 sampling locations were distributed from the shore to the center of the lake if the size allowed, and flux measurements were taken at each location and then averaged to estimate $CH_4$ or $CO_2$ fluxes from the respective lakes. This sampling at multiple locations within multiple lakes allowed us to consider the spatial variability of carbon fluxes in each cluster. Each thermokarst lake was sampled five times during the ice-free period from mid-May to mid-October of 2021 (once a month) to explore the seasonal variations of the $CH_4$ and $CO_2$ fluxes (120 lakes × 5 times). In-situ total $CH_4$ and $CO_2$ fluxes were determined between 9:00 a.m. and 18:00 p.m. using

an opaque lightweight floating chamber (diameter: 26 cm, and height: 25 cm; Supplementary Fig. 1) equipped with a closed loop to a near-infrared laser $CH_4/CO_2$ analyzer (GLA231-GGA, ABB., Canada; Precision: <0.9 ppb $CH_4$ (1 s) and 0.35 ppm $CO_2$ (1 s); Measurement rates: 0.01 to 10 Hz.)[33,58]. Specifically, the floating chamber was flushed with ambient air for ~10 s before each measurement to ensure ambient $CH_4$ and $CO_2$ concentrations inside the chamber. The chamber was then pulled into its sampling location with a rope to avoid the need to enter the lake and potentially disturb sediment and gas release. $CH_4$ and $CO_2$ concentrations in the chamber were continuously recorded for 150 s at an interval of 1 sec after an equilibration period (20 s) to eliminate any disturbance of the surface boundary layer induced by chamber deployment. At each sampling location, measurements were made for 170 sec. This measurement time (170 s) was adopted to avoid $CO_2$ accumulation in the floating chamber which could inhibit $CO_2$ emission. The 20 s equilibration period was selected based on the time needed for $CO_2$ concentration changes in the chamber to become linear. The 1 s interval was chosen to capture the change of gas concentration over time within the chamber in real time. Total $CH_4$ and $CO_2$ fluxes were calculated using the following Eq. 1[33,58]:

$$F_{total} = \frac{n_t - n_0}{A \times t} \qquad (1)$$

where $F_{total}$ is the total $CH_4$ or $CO_2$ flux (mmol m$^{-2}$ d$^{-1}$), $n_t$ and $n_0$ represent the number of moles of $CH_4$ or $CO_2$ in the chamber at time 170 and 20 s after chamber deployment (mmol), respectively, $A$ is the base area of the chamber (m$^2$), $t$ is the recording time during flux measurement (s).

Diffusion and ebullition $CH_4$ fluxes were calculated using a two-layer model to estimate their relative contributions to the total $CH_4$ release. Specifically, during flux measurements, 50 ml of surface water from a depth of 0–10 cm was collected with a 100 ml syringe (1:1 ratio of air-water). Subsequently, 50 ml of pure $N_2$ was injected into the syringe to create 50 ml of headspace. The syringe was immediately shaken for 5 min to equilibrate the headspace in the field. The headspace sample was then injected into a vacuumed airtight vial and transported to the laboratory for analysis of $CH_4$ and $CO_2$ concentrations using a gas chromatograph (Agilent 7890 A, Agilent Technologies Inc., Santa Clara, Canada). Dissolved $CH_4$ and $CO_2$ concentrations were calculated using Henry's Law adjusted for temperature and atmospheric pressure[58]. The degree of $CH_4$ and $CO_2$ saturation (S) was calculated by comparing the dissolved $CH_4$ and $CO_2$ concentration ($C_W$) with the dissolved $CH_4$ and $CO_2$ concentration at equilibrium with the local atmosphere corrected for changes in solubility ($C_{eq}$) according to Eq. 2:[33,58]

$$S = C_w / C_{eq} \qquad (2)$$

Diffusion $CH_4$ flux across the water surface into the chamber was estimated from Eq. 3:[33,58]

$$F_d = k(C_w - C_{eq}) \qquad (3)$$

where $F_d$ is the diffusion $CH_4$ flux (mmol m$^{-2}$ d$^{-1}$), $k$ is the $CH_4$ transfer coefficient (m d$^{-1}$), $C_w$ is the dissolved $CH_4$ concentration (mmol m$^{-3}$), and $C_{eq}$ is the $CH_4$ concentration in water (mmol m$^{-3}$) at equilibrium with the atmosphere in the field corrected for changes in solubility according to the Henry's law[29]. The $CH_4$ transfer coefficient ($k$) was estimated from Eq. 4:[33,58]

$$k = k_{CO_2}(S_{cCH_4}/S_{cCO_2})^{-n} \qquad (4)$$

where $k_{CO2}$ is the $CO_2$ transfer coefficient, $S_{cCH4}$ and $S_{cCO2}$ are the Schmidt number of $CH_4$ and $CO_2$, respectively, $n$ is 1/2 when wind

speeds are >3.6 m s$^{-1}$ or 2/3 if wind speeds are <3.6 m s$^{-1}$[33]. Given that $CO_2$ ebullition is negligible due to its high solubility, $CO_2$ is exclusively diffusive[59]. Due to this point, the $CO_2$ transfer coefficient was calculated from Eq. 3.

Ebullition $CH_4$ flux was determined from the difference between the measured total $CH_4$ flux and the calculated diffusion $CH_4$ flux. We would like to point out that, although the two-layer model is popularly used to evaluate $CH_4$ diffusion flux from Arctic thermokarst lakes[11,28] and alpine rivers[33], uncertainties could be introduced due to the adopted theoretical gas transfer $k$ coefficient[60]. Future studies should thus attempt to quantify $CH_4$ ebullition and diffusion fluxes from alpine thermokarst lakes using other approaches (e.g. bubble traps[61]).

During flux measurement at each cluster ($n = 30$), we quantified wind speed and atmospheric temperature with a portable anemometer (Testo 480, Testo SE & Co. KGaA, Lenzkirch, Germany). The air pressure, water temperature, oxidation-reduction potentiality, pH and dissolved oxygen concentration were measured in each thermokarst lake ($n = 120$) with a portable multiparameter water quality instrument (ProSolo Digital Water Quality Meter, Yellow Springs Instrument, Brannum Lane, USA). Atmospheric $CH_4$ and $CO_2$ concentrations were also recorded with the $CH_4/CO_2$ analyzer (GLA231-GGA, ABB., Canada) as the steady values obtained when being flushed through with ambient air. All parameters were measured five times—once a month from mid-May to mid-October, 2021.

## $^{14}$C and δ$^{13}$C isotopic analyses

To evaluate the $CH_4$ radiocarbon age and the pathway of $CH_4$ production, we quantified the $^{14}$C-CH$_4$, δ$^{13}$C-CH$_4$, and δ$^{13}$C-CO$_2$ isotopic ratios. Due to the high cost of isotopic analyses, only one thermokarst lake at each cluster was randomly sampled for $^{14}$C and δ$^{13}$C measurements. In addition, due to the limited gas samples, $^{14}$C and δ$^{13}$C isotopic signatures were measured for only 24 and 29 lakes, respectively. Specifically, a combined bubble sample was collected using 4 submerged plastic traps (diameter 0.7 m) placed along the transect from the shore to the center of each of the selected lakes during mid-July to mid-August, 2021[62]. Bubble gas from the traps was collected for about two weeks to enable sufficient volume to accumulate, and then divided into two parts. The first part was injected into 1 L pre-evacuated airtight gas-sampling aluminum bags (Dalian Delin Gas Packing Co., Ltd, China) for the determination of radiocarbon isotopic composition[63]. In particular, $CO_2$ and $H_2O$ in the sample were removed using two traps which were filled with ethanol-liquid nitrogen and Alkali lime-Magnesium perchlorate, respectively. $CH_4$ was then combusted with copper oxide to produce $CO_2$ and $H_2O$ at 950 °C. Prior to this, the copper oxide was charged with oxygen at 600 °C overnight. Following combustion, water was removed through a trap filled with Alkali lime-Magnesium perchlorate, and the pure $CO_2$ was then locked by a liquid nitrogen trap. Finally, the samples were quantified and catalytically reduced to graphite (containing ~1 mg C), and the $^{14}$C/$^{12}$C isotopic ratio was measured by accelerator mass spectrometry (0.5MV 1.5SDH-1, NEC, USA) at Third Institute of Oceanography, Ministry of Natural Resources, Xiamen, China. The measured $^{14}$C values were corrected for mass-dependent fractionation by being normalized to a fixed δ$^{13}$C value level[64] and reported as conventional radiocarbon ages (years before present, yrs BP; where 0 yrs BP = AD 1950; Supplementary Table 2).

The second part of the gas sample was stored in 20 ml glass bottles for determining the stable-carbon isotopic composition of $CH_4$ and $CO_2$[38]. Briefly, $CH_4$ in the sample was purified through a trap filled with liquid nitrogen and then combusted to $CO_2$ and $H_2O$ in an oxidation oven. After removing $H_2O$, δ$^{13}$C–CH$_4$ was measured with an isotopic ratio mass spectrometry (IRMS 20-22, SerCon, Crewe, UK) at Institute of Botany, Chinese Academy of Sciences, Beijing, China. δ$^{13}$C–CO$_2$ measurements were made using a similar procedure, but without the purification and combustion. The apparent fractionation

factor ($\alpha_C$) was calculated from $\delta^{13}C$ of $CH_4$ and $CO_2$ (‰, relative to Vienna PDB) with Eq. 5:[38]

$$\alpha_C = (\delta^{13}C-CO_2+1000)/(\delta^{13}C-CH_4+1000) \tag{5}$$

The $\alpha_C$ value indicates the dominant pathway of methanogenesis; small $\alpha_C$ values (1.04–1.055) are associated with acetate fermentation, while large $\alpha_C$ values (1.055–1.09) are caused by $CO_2$ reduction[38].

### Metagenomic sequencing, functional annotation and taxonomic analysis

To explore the potential role of the microbial community and functional genes in mediating $CH_4$ emissions from thermokarst lakes, we collected lake sediment samples (0–15 cm) at equally spaced intervals along the transect from the shore to the center of each lake during mid-July to mid-August, 2021. The sediment samples were passed through a 2 mm sieve with visible roots being removed. Due to the high experimental cost, only one lake at each of the 30 clusters was sampled for metagenomic analysis. Lake sediment samples were immediately transported to the laboratory and stored at −20 °C until the metagenomic analysis. Total DNA was extracted from thawed sediment samples (0.4 g) using the DNeasy PowerSoil kit (Qiagen, Hilden, Germany) according to the manufacturers' instructions. The ratios of spectrometry absorbance for the extracted DNA were between 1.7 and 1.9 at 260/280 nm and between 1.7 and 2.1 at 260/230 nm.

The metagenomic sequencing was carried out using the DNBSEQ-T7 platform (BGI, Shenzhen, China) to obtain $2 \times 150$ bp paired-end reads. Raw reads and adapters were first removed using SOAPnuke software (-n 0.001 -l 20 -q 0.4 --adaMR 0.25 --adaMis 3 --outQualSys 1) to generate trimmed reads consisting of approximately 15–18 Gb of sequencing data for each sample. Clean reads for each sample were assembled using Megahit (v.1.2.9)[65], and contigs with length > 300 bp were retained. Then, open reading frames (ORF) were predicted by Prodigal (v.2.6.3)[66], and dereplicated with 95% nucleotide identity using CD-HIT-EST (v.4.8.1)[67]. The translated proteins of non-redundant gene clusters were annotated by searching against the EggNOG (v.5.0) database[68] using DIAMOND (v.2.0.9)[69]. The final annotation table was obtained from the KEGG database[70] which is derived from EggNOG. To estimate the relative abundance of representative non-redundant genes in each sample, Salmon (v.1.5.1)[71] was used to calculate the TPM (transcripts per million) of each predicted gene via mapping to clean paired reads of each sample. Taxanomic annotation was evaluated using Kraken2 (v.2.1.2)[72] against the mini database (downloaded on Feb.–2022), and then visualized using GraPhlAn (v.0.9.7)[73].

To further verify the annotations of genes involved in $CH_4$ production, assembled contigs with length ≥1000 bp were used for identifying the methyl-CoM reductase alpha subunit (*mcrA*). Specifically, contigs with length ≥1000 bp were predicted using Prodigal (v.2.6.3) and then screened using HMM profiles (PF02249 and PF02745) derived from the Pfam database[74] with hmmsearch (HMMER v.3.3.2)[75]. Putative *mcrA* genes were validated by phylogenetic tree with reference sequences derived from Annotree[76] based on HMM profiles (PF02249 and PF02745 from Pfam) and Zhou et al.[77]. Putative *mcrA* proteins were aligned with reference sequences using MUSCLE (v.3.8.31)[78] and built for a phylogenetic tree using FastTree (v.2.1.11)[79] with the WAG + GAMMA models. Validated *mcrA* genes were mapped to clean paired reads of each sample by Salmon (v.1.5.1) to obtain counts, lengths and effective lengths. Counts per gene were normalized to reads per kilobase per million mapped reads (RPKM).

### Statistical analyses

We conducted a series of statistical analyses to explore the basic characteristics of $CH_4$ emissions from alpine thermokarst lakes. Specifically, one-way ANOVAs were carried out to evaluate the differences in total carbon emissions ($CH_4 + CO_2$), the contribution of ebullition to total $CH_4$ fluxes, $\alpha_C$, the relative abundances of methanogenic functional genes and community from thermokarst lakes located in the three grassland types (alpine steppe, alpine meadow and swamp meadow). Tukey's HSD difference test was used for multiple comparison at a significance level of $\alpha = 0.05$. $p$ value corrections were performed for multiple comparisons using the Benjamini–Hochberg correction factor[80]. Regression analyses were performed to examine the relationship of the ebullition proportion to total $CH_4$ fluxes with elevation and air pressure. Before applying regression analyses, we conducted outlier analysis for the proportion of ebullition to total $CH_4$ fluxes based on Boxplot Procedures[81]. Log transformation was conducted when the continuous variables and their residuals violated assumptions of normality.

We upscaled $CH_4$ and $CO_2$ fluxes from lake levels to the regional scale using Monte Carlo analysis that ran 1000 iterations for each of the grassland types (including alpine steppe, alpine meadow and swamp meadow) where thermokarst lakes are mainly located. Each iteration randomly resampled a $CH_4$ or $CO_2$ flux (for one of the three grassland types) based on a normal distribution surrounding the mean and standard deviation. Then, to generate the total thermokarst lake $CH_4$ or $CO_2$ flux per unit of time for each grassland type, we multiplied the randomly resampled $CH_4$ or $CO_2$ flux values by the area of thermokarst lakes for each grassland type which was determined by the distribution of thermokarst lakes on the Tibetan Plateau[17] and the vegetation map was derived from China's Vegetation Atlas[82]. Subsequently, we multiplied the total $CH_4$ or $CO_2$ flux per unit of time by the ice-free season duration (~200 d) to obtain the $CH_4$ or $CO_2$ emissions from each grassland type. Finally, we summed $CH_4$ or $CO_2$ emissions from each grassland type to estimate total $CH_4$ or $CO_2$ emissions. All statistical analyses were performed using R statistical software v3.6.2 (http://r-project.org)[83].

### Reporting summary

Further information on research design is available in the Nature Portfolio Reporting Summary linked to this article.

## Data availability

All data supporting the findings are available in the Figshare data repository (https://doi.org/10.6084/m9.figshare.22743968)[84] and Supplementary Information. The nucleotide sequences generated by metagenome sequencing have been deposited in the NCBI database (ncbi.nlm.nih.gov/bioproject/?term=PRJNA942440).

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

## Acknowledgements

We thank Dr. Liwei Zhang (College of Urban and Environmental Sciences, Peking University) for valuable advice on flux measurement and calculation, Dr. Pengfei Liu (Center for the Pan-Third Pole Environment, Lanzhou University) for helpful suggestions regarding bioinformatic analyses and data interpretations, and Dr. Katey Walter Anthony (Water and Environmental Research Centre, University of Alaska Fairbanks) for constructive comments on an early version of the manuscript. Permissions to work and collect gas and sediment samples across the study area were granted by the Three-River-Source National Park Management Bureau. This work was supported by the National Natural Science Foundation of China (31988102 and 31825006, Y.Y., and 32201359, G.Y.), National Key Research and Development Program of China (2022YFF0801903, Y.Y.), and China National Postdoctoral Program for Innovative Talents (BX20200363, G.Y.).

## Author contributions

Y.Y. and G.Y. designed this research. G.Y. and Z.Z. performed field samplings. G.Y., Z.Z., L.K. and S.Q. performed laboratory experiments. G.Y., Y.S., L.K. and Y.P. analysed data. G.Y., B.A., D.O., C.K. and Y.Y. wrote the manuscript with input from other co-authors.

## Competing interests

The authors declare no competing interests.
