## [Peer Review File · Nature Communications]

Methane emissions from alpine thermokarst lakes on the Tibetan Plateau: amount, origin, and methanogenic microorganismsReviewer #1 (Remarks to the Author):

See attached

Reviewer #1 Attachment on the following page

In the submitted manuscript, ‘Vital contribution of methane to total carbon emissions from thermokarst lakes’ by Yang et al., carbon emission was measured in 120 thermokarst lakes in the Tibetan Plateau during the open water season. Carbon emissions in thermokarst lakes have received increasing attention in recent years, as the formation and transformation of such lakes are sensitive to warming temperatures, yet, information on emissions from the Tibetan Plateau is limited compared to high-latitude thermokarst lakes. Therefore, this study is unique in that it includes carbon emission from an understudied thermokarst lake region, and is timely given their climate sensitivity, with results that have important implications for understanding carbon emissions from lakes globally.

Main Points:

Overall the paper is well written and structured. There are a few points in the methods that need to be clarified and/or expanded.

- 1) The sampling scheme is a bit unclear, i.e. 120 lakes were sampled in total (30 clusters with 4 lakes in different size class) on five sampling occasions. Was each lake sampled 5 times (i.e. $120 \times 5 = 600$ samples) or was each lake visited 1 time during one of the 5 sampling occasions. Specific sections in the text where this needs to be clarified are mentioned below.
- 2) From the methods it seems that CO₂ concentration was also measured but this is not reported in the paper, i.e. only CO₂ flux is reported. Further in making the upscaling emission estimates only CH₄ is upscaled. Is there a reason for this? It could be interesting to also include CO₂ emission estimate, as data from this region is often lacking in global lake CO₂ upscaling efforts.
- 3) Emission estimates are only made for the open water period, which as stated make the emissions conservative. Is there any information on CH₄/CO₂ concentrations below ice and if so is there enough to make a rough back of the envelope calculation of the contribution of the ice-cover period to annual emissions? This is just a suggestion and if adequate data doesn't exist to make such estimates this could be mentioned.

General Comments:

Title:

Tibetam Plateau could be mentioned in the title as one of the main findings was that these thermokarst lakes showed different results from thermokarst lakes at higher latitudes. e.g. “Vital contribution of methane to total carbon emissions from thermokarst lakes in the Tibetan Plateau”

Abstract

L27: region in the world

L 38-41: Is there really a general view that CH₄ compared to CO₂ is of minor importance in global lakes? It perhaps depends on what region you are studying, although CH₄ is lower, as mentioned the GWP potential is higher and thus it is significant in terms of CO₂.

Introduction:

L 78: Since this is a hypothesis could rewrite as, “CH₄ emissions from thermokarst lakes in this region have a high contribution to total carbon emissions.”

L 81-85: As has been done in the methods, good to point out here that there are 4 lakes in each cluster (120 lakes / 30 clusters). Is each lake sampled 5 times or is it that sampling took place between mid-May to mid-Oct and each lake was sampled once?

L 87 – 92: Currently % supersaturation is mentioned first and then concentrations. Could instead have concentrations first and then % supersaturation. Since L 86 states concentration was measured, you are expecting to have the concentration reported and then put in the context of supersaturation.

L90: delete “respectively”

L93: compared to CH concentration

L101: and even higher than Arctic

L112: this result opposes previous studies from Arctic

L132: Was CO₂ concentration measured or only flux? If measured could be nice to include.

L147: Is the global lake (1.1-6.4%) referring to the contribution of CH₄ to total carbon emissions (i.e. 7.3% in this study) or CO₂ equivalent emission (i.e. 44.6% in this study)?

L183-184: There seems to be a word missing from the first part of this sentence

L193: delete consumption (CH₄ oxidation is the same thing)

L210-212: This sentence is confusing and could be simplified. E.g. decrease in O₂ may increase CH₄ production and keep CH₄ produced from being oxidized and thus increase the CH₄/CO₂ ratio. CH₄ release is mentioned here and I am wondering if the authors have explored how low atmospheric pressure impacts CH₄ ebullition, this could be an interesting point to mention here.

L 216: See comment L132, if CO₂ concentrations were measured it would be interesting to know if they compare to lakes with similar pH.

L218: Remove “Considering that” and end sentence after (Fig 5).

L230: remove these

L249: locating → located

L251: but still have the potential to contribute to CH₄ emissions

L 247-251: Are there estimates of thermokarst lake expansion/loss in this region? In addition to the 20% non-grassland lakes not accounted for, there are likely new lakes.

L 256: Could consider making a back of the envelope calculation for ice-covered period. Is there any data from below ice for lakes in this region that could be used for such an estimate? In Denfeld et al. 2018 (doi: 10.1002/lol2.10079), Figure 5, shows %CH₄ ice cover for a global set of lakes. Could consider using the average value, however, none of the lakes in that study are from this region, so perhaps not really relevant to use.

L 265: First study ever or in this region?

Figures:

Fig 1: What are the lines in Fig 1B?

Fig 1 and Fig 2: It seems there are 30 data points and thus the figure is showing the flux for each cluster, is this correct? Please clarify in the figure legend.

Methods:

L457: The Tibetan Plateau, the largest alpine permafrost region in the world, has an average elevation over 4000 m above sea level.

L466: So is the region snow/ice covered from mid Oct to mid-May? Could mention the average number of days snow/ice covered, if known.

L 480-488: See comment L 81 above. So in each cluster 4 lakes were selected = 120 lakes.

L 488-490: “increased the probability of capturing ebullition” makes it sound like the bubbles were targeted and not randomly captured. Rather by taking spatial variability into account diffusive flux and ebullition were both accounted for, making the CH₄ emission estimates robust.

L492-500: so there were 130 readings (150 s – 20 s equilibration, 1 per s)? What is the sensitivity of the analyzer, i.e. 1 sec readings seem rather quick. But it seems flux is calc from two sampling times, i.e. 20 and 170 s, so over a 2.5 min period. This could be clarified here. Why was this amount of time chosen?

L505: CH₄ concentrations in the water.

L507: What was the ratio of air-water for the headspace extraction? Was the headspace extracted in the field? If using the same exact method as reference 56 state that so that the reader knows they can look into the details there.

L507-509: See L132 above. It seems CO₂ concentration was measured.

L 522: How was wind speed measured to determine the Sc?

L 528: Was each 120 lakes sampled once or multiple times over the open water season? Please clarify.

L567-568: Could consider putting this info at the start of the section (L 536). How was the one lake per cluster selected, i.e. random or a certain size class?

L574: we collected lake sediment samples

L609: Could the same calculation be made for CO₂?

Extended Data Figure 1: The total number for each grassland type is reported in the caption, summing to 120, but it is unclear if this is the total for all lakes or for each sampling date (i.e. does each bar in the box plot contain 120 points)?

Extended Data Figure 2: If I understand correctly each bar is n=30, so does that mean the average CH₄ and CO₂ flux from each cluster is reported for each data point in this graph?

Extended Data Table 1: The area → Lake area in table heading

Reviewer #2 (Remarks to the Author):

This paper investigate the CH₄ (and partly CO₂) emission from thermokarst lakes located at the Tibet plateau. The region hold a large part of the alpine permafrost globally, and experience changes in permafrost dynamics that could potentially release substantial amounts of carbon to the atmosphere, directly from land or from downstream aquatic systems. The carbon emission from thermokarst lakes of the region is poorly constrained. Thus, considering that my concerns can be adequately addressed, the study is novel and important for the larger scientific community focusing on carbon cycling and climate change.

In general I find the paper clear and well structured, but I also see the need of clarifying some parts and also changing the aim and focus of result presentation and discussion. Below I first summarize main comments followed by a set of more detailed comments. My main comment is the large focus on the CH₄:CO₂ balance and comparing CH₄ emission (not CO₂) with the CH₄ exchange of terrestrial systems. In order to understand the role of the lakes in the overall C balance of the region you need to focus on the absolute C emission, and compare this with the net C exchange of the terrestrial systems.

I also have a major comment regarding the collection of data. The sampling program is impressive and ensure that the data collected is fairly representative for the carbon emission from thermokarst lakes of the region. They sampled 120 lakes of different size distributed along an extensive (1100km) transect. Sampling was carried out monthly (5 times) during the open water season using established methods. However, additional details (see below) is needed to ensure that the sampling did not introduce and bias in the results.

Detailed comments (with reference to line numbers):

L74-76. In the end it is not the flux proportion, but rather the absolute amount released to atm that matter. Focusing on the proportion can give wrong impression since systems with very low total emission can have high CH₄:CO₂ ratios (if CO₂ emissions are low).

L78-79. I suggest to skip hypothesis 2. It is included in hypothesis 1, and hypothesis 1 is stronger since it focus on total emission (not relative).

L89-92. Good to give the gas concentrations but i suggest to not continue and make interpretations of the concentration patters. It is the gas fluxes that are important. If you still choose to discuss the concentration it would be reasonable to also discuss gas exchange velocity (k) in order to understand their individual and combined effect on gas fluxes.

L114- The discussion and conclusion based on the "2 aspects" are not fully clear. If Tibetan soils have generally younger surface soils, but deeper active layers, compared to arctic soils, would not these 2 factors work in different directions and result in less difference in age of released C?

L117. Not obvious how deep active layer would result in less disturbance of deep soils

L118-119. This part is not clear. Do you mean low contribution of potential release of C as a result of further deepening of active layer?

L132. Please rephrase. You did not measure CO₂ fluxes, they were calculated from concentration and k data (derived from CH₄ flux and concentration measurements).

L142-143. This part need additional explanation. GPP in water could be regarded to still be reasonably captured by its general effect on pCO₂ in water (and thus on atm fluxes). Or do you mean that there were emergent plants of which CO₂ uptake were not included in chamber estimates?

L146-147. The difference between your lakes (7.3%) and global lakes (1.1-6.4%) are

rather small. Given uncertainties in data i suggest to not make a major story of this difference. A tendency for high relative emission of CH₄ vs CO₂ can be noted, but should not be the main focus of the paper. As mentioned earlier the focus should be on absolute C emissions.

L149-216. This part is very long and focused on factors potentiality causing high fractional CH₄. Yet to understand the high ratio you also need to discuss factors that could result in relatively low CO₂.

L163. As written it is not clear if this is different compared to other lakes, e.g. to global and arctic permafrost lakes? This needs to be shown in order to propose this as an explanation for high relative proportion of CH₄ to C emission.

L218-223. Long and complicated sentence. Please revise.

L229. It is not clear if you mean 1.4-8.4 fold greater than the total or the relative contribution.

L237. "Expressed differently.." is maybe not the best wording since this sentence compare with other ecosystem types. Replace with "Further,.."?

L239-241. This statement is not supported by the comparison in 2 previous sentences, e.g. that lake CH₄ emissions offset 15% of CH₄ sink of steppe and meadows (that cover 95% of grasslands), and roughly equal emissions from swamps (which cover only 5% of grasslands).

Further, do make this landscape scale comparison relevant and allow assessment of overall importance of the lakes for the larger C balance you should also include CO₂ exchange (of land and water) in these comparisons.

L251. Provide reference to support this statement about that these lakes are contributors to emissions.

L253-255. Are the lakes deep enough to allow significant gas accumulation under ice? What are the mean depth of these lakes and the thickness of ice in winter? Fig 1 suggest that the lakes are very shallow and would not allow significant gas accumulation in winter. Also, you have one reference for statement of particularly high CH₄ release following ice-out. Good to refine statement and include other references that do not show clear support of this (see compilation by Denfeld et al. 2018 L&O letters, doi: 10.1002/lol2.10079). Also, the lakes studied by Walter et al are likely quite different compared to your lakes (depth, permafrost conditions...).

L256. Based on past comment i am not convinced this is true, and I suggest to not include it. Including ice-off period could however increase absolute C emission by both CO₂ and CH₄, so this part can be revised accordingly to reflect this.

L274-275. The absolute C emission is more relevant than the ratio. There are lakes with CH₄ release but CO₂ uptake which with this reasoning would suggest that they are very important despite they may be negligible in terms of overall C emissions. I would downplay this argumentation.

L476-. Give more details to ensure nonbiased data. How were the chambers sampled to avoid disturbance, which is easily created in small shallow systems (where turbulence enhance fluxes).

L492-493. How were CH₄ collected? In situ sensor or manually from boat? The later can create turbulence and bias flux estimates so important that you clarify hoe it was done..

L525. Did you measure atm CO₂ to be used in calculations? Please provide details.

L610. Did you not also upscale CO2 fluxes?

Reviewer #3 (Remarks to the Author):

Review on « Vital contribution of methane to total emission from thermokarst lakes »

The manuscript "Vital contribution of methane to total carbon emissions from thermokarst lakes" by Yang et al., presents measurement of CH₄ and CO₂ fluxes from lakes located on the Tibetan plateau. These data are highly valuable and will be of interest to the community. However, the manuscript suffers from lots of imprecisions, both in the data interpretation and reference to the literature. There are a few examples (listed below) where the interpretation given by the authors does not match the presented results. The authors also generally oversimplify the literature and compare their study with "the arctic", when there is obviously no unique behavior of high latitude lakes, but a wide diversity of geologic and climatic contexts.

Major comments:

- **Ebullition vs diffusion:** A major result emphasized in this manuscript is the high contribution of ebullition fluxes for CH₄. Yet, this partitioning is based on the use of a theoretical k value. This implies a huge uncertainty in the partitioning between the two pathways of CH₄ emission. The ebullition fluxes could have been identified looking at the continuous records of [CH₄] in the chamber. The authors should include examples of fluxes that have been attributed to ebullition and diffusion
- **Origin of emitted CH₄.** The conclusion stated by the authors : "The measured CH₄ radiocarbon ages were in a wide range between modern and 3,810 years before present (yrs BP; n = 24; Fig.2A and Supplementary Table 2), indicating that CH₄ production was mainly from recently fixed organic carbon" does not match the results. Yet, 3810 BP is not really "recently fixed organic carbon". When the authors cite an average age of 6100 yrs BP for Tibetan soils, does it mean that those soils (frozen? we do not know) contribute to half of the emitted carbon in form of CH₄?
- **The "Arctic"** . The authors present an over-simplistic view, comparing the Tibetan plateau with the "Arctic", as if the Arctic was a uniform entity. This is especially striking when the authors cite a unique AL depth (0.7) and an average age of 14,000 +/- 4000 yrs BP for emitted CH₄ from the Arctic.

Detailed comments

- L. 34-37. This long sentence could be split.
- L. 81. "120 lakes in 30 clusters" : The authors could precise what was the timing of the measurements (time in the season, time of the day), and how long did a campaign last?
- L. 100-102. Comparison with Wik et al., 2016. In this publication the high range corresponds to the measured values in this study.
- L. 127. Supplementary Figure 1 presents "Seasonal and regional patterns of CH₄ concentrations from thermokarst lakes in Tibetan alpine permafrost region". Not the age of organic carbon in soils
- L. 155. Is -72.5 +/- 1.1 the mean +/- standard deviation? On the plot, d13C-CH₄ range from -32 to -85, it seems that the dispersion is much larger?
- L. 156. Do the authors have any comments on the d13C-CO₂ value dispersion?
- L. 184-186. This statement is not supported by any reference. Please include one.
- L. 187. Any more recent/ adequate reference for H₄ oxidation in high latitude lakes?
- L. 191 . Any reference for the very general statement on methanotroph abundance in the Arctic?
- L.202. Interesting discussion, but the authors should strengthen their demonstration of the contribution of ebullition fluxes.
- L. 208-212. Unclear statement
- L. 233. In the map published by Wei et al., All lakes in a permafrost area are considered to be thaw lakes. Do the authors have any comment on this statement? What

are the implication in terms of CH₄ fluxes?

Responses to Reviewer #1

[Comment 1] In the submitted manuscript, ‘Vital contribution of methane to total carbon emissions from thermokarst lakes’ by Yang et al., carbon emission was measured in 120 thermokarst lakes in the Tibetan Plateau during the open water season. Carbon emissions in thermokarst lakes have received increasing attention in recent years, as the formation and transformation of such lakes are sensitive to warming temperatures, yet, information on emissions from the Tibetan Plateau is limited compared to high-latitude thermokarst lakes. Therefore, this study is unique in that it includes carbon emission from an understudied thermokarst lake region, and is timely given their climate sensitivity, with results that have important implications for understanding carbon emissions from lakes globally.

[Response] Many thanks for the reviewer’s excellent comments. These comments listed below help us to conduct a thorough revision on the manuscript. We really appreciate this professional review which greatly improved our paper. Thank you! Detailed modifications please see our responses to the following comments.

Major comments:

[Comment 2] Overall the paper is well written and structured. There are a few points in the methods that need to be clarified and/or expanded.

1) The sampling scheme is a bit unclear, i.e. 120 lakes were sampled in total (30 clusters with 4 lakes in different size class) on five sampling occasions. Was each lake sampled 5 times (i.e. $120 \times 5 = 600$ samples) or was each lake visited 1 time during one of the 5 sampling occasions. Specific sections in the text where this needs to be clarified are mentioned below.

[Response] Sorry for the poor description about the sampling design in previous MS. In this study, 30 clusters were selected along a 1,100 km transect across the Tibetan alpine permafrost region (**step 1; Fig. R1a**). At each cluster, four thermokarst lakes were selected to cover different lake sizes (Fig. R1b). **Total 120 thermokarst lakes were thus sampled (30 clusters × 4 lakes) (step 2; Fig. R1b). Each thermokarst lake was sampled 5 times during the ice-free period from mid-May to mid-October of 2021 (once a month) to explore seasonal variation of CH₄ or CO₂ flux (120 lakes × 5 times = 600 samples; step 3; Fig. R1d-h).** During each measurement, 4 to 6 sampling locations were selected from the shore to center of the lake if the size allowed

(Fig. R1c), and their flux measurements were taken and **averaged** to estimate CH₄ or CO₂ flux from the respective lakes (**Step 2; Fig. R1c**). To clarify these points, we have rephrased the sampling method (Page 4, lines 76-81 and Pages 16-17, lines 354-380) and added a schematic diagram in the revised MS (Fig. R1; Page 34, lines 767-783 and Page 38, lines 831-832).

Step 1 Selecting sampling clusters: 30 clusters were located in three representative permafrost regions (Madoi, Budongquan-Nagqu-Zadoi and Qilian sections)

Step 2 Sampling at each cluster: Four thermokarst lakes were selected to cover different lake sizes at each cluster. In each lake, 4 to 6 sampling locations were distributed from the shore to the center.

Step 3 Repeated sampling at each cluster during the ice-free period: Each lake was sampled five times at monthly intervals during the ice-free period from mid-May to mid-October of 2021 to explore seasonal variations of CH₄ and CO₂ fluxes.

Fig. R1 The flow chart of sampling campaign. Our field sampling consisted of the following three key steps. First, we chose 30 clusters of thermokarst lakes along a 1,100 km transect on the Tibetan Plateau (a). Second, multiple locations within multiple lakes were selected at each cluster to eliminate spatial heterogeneity. In particular, four thermokarst lakes with different sizes were selected at each cluster (b). In each lake, 4 to 6 sampling locations were distributed from the shore to center (c), and their flux measurements were taken and averaged to estimate CH₄ or CO₂ flux from the respective lakes. Finally, each lake was sampled five times at monthly intervals during the ice-free period from mid-May to mid-October of 2021 to explore seasonal variations of CH₄ and CO₂ fluxes (d-h). *In-situ* CH₄ and CO₂ fluxes were determined using an opaque lightweight floating chamber equipped with a closed loop to a near-infrared laser CH₄/CO₂ analyzer (GLA231-GGA, ABB., Canada). In panel (a), the permafrost map of the Northern Hemisphere was obtained from the National Snow & Ice Data Center (Brown et al. 1998), while the spatial distribution of permafrost on the Tibetan Plateau was derived from Zou et al. (2017). Three circles indicate three representative permafrost regions across our study area, including the Madoi, Budongquan-Nagqu-Zadoi and Qilian sections. Photos were taken by Guibiao Yang.

[Comment 3] 2) From the methods it seems that CO₂ concentration was also measured but this is not reported in the paper, i.e. only CO₂ flux is reported. Further in making the upscaling emission estimates only CH₄ is upscaled. Is there a reason for this? It could be interesting to also include CO₂ emission estimate, as data from this region is often lacking in global lake CO₂ upscaling efforts.

[Response] Very good comment! We did measure CO₂ concentration and estimated regional CO₂ emission from thermokarst lakes on the Tibetan Plateau. However, we reported little CO₂ characteristics in the original MS because previous version mainly emphasized CH₄ emissions. Nevertheless, as mentioned by this reviewer that the data of CO₂ emission from alpine thermokarst lakes on the Tibetan Plateau are lacking, combining the suggestion from the second reviewer (**Comment 18: assessment of overall importance of the lakes for the larger C balance you should also include CO₂**), **we have added more descriptions for CO₂ concentrations, saturations and fluxes, and also upscaled CO₂ fluxes using the same approach as CH₄ in the Supplementary materials of revised MS as follows:** *‘Both CO₂ concentrations and fluxes were determined simultaneously with CH₄ measurements. These measurements revealed that dissolved CO₂ concentrations did not have obvious seasonal differences, but varied between 16.0 and 34.7 μmol L⁻¹ across the 30 sampling clusters with a mean value of 19.3 ± 0.7 μmol L⁻¹ (n = 30; Supplementary Fig. 3b). 62% of the studied thermokarst lakes were supersaturated in CO₂ with respect to the local atmosphere (ranging from 89.9 to 206.4% with an average of 108.5 ± 3.9%; n = 30). CO₂ fluxes showed high spatial variability across the 30 clusters, ranging from -33.7 to 445.0 mmol m⁻² d⁻¹ with an average value of 170.4 ± 21.8 mmol m⁻² d⁻¹ (n = 30; Supplementary Fig. 9). Thermokarst lakes distributed in alpine meadow and swamp meadow had higher CO₂ fluxes than those located in alpine steppe (Supplementary Fig. 4c). However, no significant difference was detected between thermokarst lakes located in alpine meadow and swamp meadow. As a whole, we observed high CO₂ flux in alpine thermokarst lakes, with values being ~5-times greater than those (35.2 mmol m⁻² d⁻¹) from small ponds (size class < 0.001 km²) (Holgerson and Raymond 2016). To obtain the regional estimation, we upscaled CO₂ fluxes from the lake level to the regional scale using the same approach as CH₄. The upscaling analyses demonstrated that CO₂ emissions from alpine thermokarst lakes on the Tibetan Plateau were 2084.7 Gg (10⁹ g) CO₂ yr⁻¹ (Table 1). Overall, despite emergent plants and their associated CO₂ uptake,*

alpine thermokarst lakes are still expected to a significant carbon source due to the sparsity of plants in most lakes.’ (Page 2, lines 5-25 in the Supplementary Note 1). Notably, to improve the readability of the manuscript, we have added the relevant descriptions about CO₂ concentrations and fluxes in the Supplementary materials of the revised MS. Thanks for your understanding!

[Comment 4] 3) Emission estimates are only made for the open water period, which as stated make the emissions conservative. Is there any information on CH₄/CO₂ concentrations below ice and if so is there enough to make a rough back of the envelope calculation of the contribution of the ice-cover period to annual emissions? This is just a suggestion and if adequate data doesn’t exist to make such estimates this could be mentioned.

[Response] Very good suggestion! We agree with the reviewer that our estimation would be conservative without considering CH₄ and CO₂ emissions during the ice-cover period. Nevertheless, unfortunately, CH₄ and CO₂ concentrations and fluxes during the ice-cover period are not monitored in this study due to the rugged environment, such as extremely low temperature, oxygen limitation, and traffic inconvenience on the Tibetan Plateau. Moreover, current measurements mainly came from the high-latitude region (Denfeld et al. 2018), with the corresponding measurements from high-altitude permafrost region being scarce. Due to this situation, we made a rough estimation based on those observations from high-latitude region. Specifically, **using the average annual contribution of carbon emissions from high-latitude lakes at ice-cover (CO₂: 17%; CH₄: 27%), CH₄ and CO₂ emissions from alpine thermokarst lakes on the Tibetan Plateau during the ice-cover period could be estimated to be 427.0 Gg CO₂ yr⁻¹ and 28.3 Gg CH₄ yr⁻¹, respectively.** Given the annual contribution of carbon emissions during ice-cover period observed from the high-latitude lakes may not be simply applied to thermokarst lakes in high-altitude permafrost region, more measurements during the ice-cover period are needed to integrate carbon budgets in alpine thermokarst lakes on the Tibetan Plateau. **We have clearly stated this point in the revised MS as follows:** *‘Although this study advances our understanding of CH₄ release from thermokarst lakes on the Tibetan Plateau, it does have some limitations. First, carbon emissions during the ice-cover period were not considered in this study. To make a rough evaluation of ice-cover carbon emission, we applied the average*

percent contribution of ice-cover to annual carbon emissions from high-latitude lakes (CO_2 : 17%; CH_4 : 27%) (Denfeld et al. 2018). These increased annual emissions estimate from alpine thermokarst lakes on the Tibetan Plateau to 427.0 Gg CO_2 yr⁻¹ and 28.3 Gg CH_4 yr⁻¹, respectively. Nevertheless, given the potential differences between high-latitude and high-altitude permafrost regions, such as climate, thermokarst lake depth and ice-cover duration (Wang et al. 2022), the annual contribution of carbon emissions during the ice-cover period observed from high-latitude lakes may not be simply applied to thermokarst lakes in high-altitude permafrost region. More measurements during the ice-cover period are thus needed to further advance our understanding of CH_4 emissions in alpine thermokarst lakes.’ (Page 13, lines 278-290).

Minor comments:

[**Comment 5**] **Title:** *Tibetan Plateau could be mentioned in the title as one of the main findings was that these thermokarst lakes showed different results from thermokarst lakes at higher latitudes. e.g. “Vital contribution of methane to total carbon emissions from thermokarst lakes in the Tibetan Plateau”*

[**Response**] Combining suggestions from this reviewer and the second reviewer (**Comment 22:** *The absolute C emission is more relevant than the ratio*), we have rephrased the title as follows: ‘Methane emissions from alpine thermokarst lakes on the Tibetan Plateau: amount, origin and methanogenic microorganisms’ (Page 1, lines 1-2).

[**Comment 6**] **Abstract:** *L27: region in the world*

[**Response**] To meet the criteria of *Nature Communications* (Abstract should be approximately 150 words), we have refined this sentence and deleted these words in the revised MS. Thanks for your understanding!

[**Comment 7**] *L38-41: Is there really a general view that CH_4 compared to CO_2 is of minor importance in global lakes? It perhaps depends on what region you are studying, although CH_4 is lower, as mentioned the GWP potential is higher and thus it is significant in terms of CO_2 .*

[**Response**] We agree with this reviewer’s opinion that the relative proportion of CH_4

vs. CO₂ flux depends on the study region, and is higher if considered in terms of CO₂-equivalent emissions. For instance, Serikova et al. (2019) reported that CH₄ flux from thermokarst lakes of Western Siberia could contribute to 12% of total carbon flux, and its contribution to the total carbon emission were up to 58.1% in terms of the GWP, which was greater than our value obtained in this study (44.6%). **This result suggests that original view** (*CH₄ compared to CO₂ is of minor importance in global lakes*) **may not be general**. Following this point raised by this reviewer together with suggestion from the second reviewer (**Comment 22:** *The absolute C emission is more relevant than the ratio*), we have downplayed the argumentation about the flux proportion, **focused on absolute carbon emission, and thus rephrased this sentence in the revised MS** as follows: ‘Overall, multiple parameters obtained in this study provide benchmarks for better predicting the strength of the feedback between permafrost carbon and climate.’ (Page 2, lines 32-34). Thanks for your understanding!

[Comment 8] L78: Since this is a hypothesis could rewrite as, “CH₄ emissions from thermokarst lakes in this region have a high contribution to total carbon emissions.”

[Response] Combining suggestions from this reviewer and the second reviewer (**Comment 5:** *I suggest to skip hypothesis 2. It is included in hypothesis 1, and hypothesis 1 is stronger since it focuses on total emission*), **we have downplayed this argument about the flux proportion, and rewritten the corresponding hypothesis as follows:** ‘Thermokarst lakes in this permafrost region are thus expected to behave as significant hot spots for CH₄ emissions.’ (Page 4, lines 66-67).

[Comment 9] L81-85: As has been done in the methods, good to point out here that there are 4 lakes in each cluster (120 lakes / 30 clusters). Is each lake sampled 5 times or is it that sampling took place between mid-May to mid-Oct and each lake was sampled once?

[Response] Following the reviewer’s comments, we have clearly stated that we conducted a large-scale sampling campaign across **120 thermokarst lakes in 30 clusters (four lakes in each cluster: 4 lakes/cluster × 30 clusters)** along a 1,100 km transect on the Tibetan Plateau. We have also clearly stated that **each lake was sampled five times at monthly intervals during the ice-free period from mid-May**

to mid-October of 2021 (Fig. R1; Page 4, lines 76-80).

[Comment 10] L87 – 92: Currently % supersaturation is mentioned first and then concentrations. Could instead have concentrations first and then % supersaturation. Since L86 states concentration was measured, you are expecting to have the concentration reported and then put in the context of supersaturation.

[Response] We agree with the reviewer’s opinion about the order of concentrations and saturation, and have rephrased these sentences as follows: ‘Across the 30 sampled clusters, CH₄ concentrations ranged from 107.1 to 159.4 nmol L⁻¹ with a mean of 136 ± 30 nmol L⁻¹ (n = 30; Supplementary Fig. 2a). CH₄ was supersaturated relative to the local atmosphere in all the studied thermokarst lakes across the clusters with a mean value of 2,921 ± 62% (ranging from 2,393 to 3,719%; n = 30).’ (Page 5, lines 94-98).

[Comment 11] L90: delete “respectively”

[Response] Done as suggested.

[Comment 12] L93: compared to CH₄ concentration

[Response] Combining this with the second reviewer’s comment (Comment 6: L89-92. Good to give the gas concentrations but I suggest to not continue and make interpretations of the concentration patterns.), we have deleted this sentence in the revised MS. Thanks for your understanding!

[Comment 13] L101: and even higher than Arctic

[Response] Combining this with the third reviewer’s comment (Comment 7: Comparison with Wik et al., 2016. In this publication the high range corresponds to the measured values in this study.), we have revised this sentence as follows: ‘The mean CH₄ flux during the ice-free season was 13.4 ± 1.5 mmol m⁻² d⁻¹. This value is at the high end of the range reported from Arctic thermokarst water bodies regarded as hotspots for CH₄ release (Wik et al. 2016; Kuhn et al. 2021)’ (Page 6, lines 108-110).

[Comment 14] L112: this result opposes previous studies from Arctic

[Response] Combining this with the third reviewer’s comment (Comment 4: The authors present an over-simplistic view, comparing the Tibetan plateau with the

“Arctic”, as if the Arctic was a uniform entity.), we have deleted this sentence about the comparison of thermokarst lakes between the Tibetan Plateau and arctic region. Thanks for your understanding!

[Comment 15] L132: Was CO₂ concentration measured or only flux? If measured could be nice to include.

[Response] Yes, as mentioned above, both CO₂ concentration and fluxes were measured in this study (Fig. R2). Following the reviewer’s comment, **we have added more descriptions about CO₂ concentrations and fluxes as follows**: ‘Both CO₂ concentrations and fluxes were determined simultaneously with CH₄ measurements. These measurements revealed that dissolved CO₂ concentrations did not have obvious seasonal differences, but varied between 16.0 and 34.7 μmol L⁻¹ across the 30 sampling clusters with a mean value of 19.3 ± 0.7 μmol L⁻¹ (n = 30; Supplementary Fig. 3b). 62% of the studied thermokarst lakes were supersaturated in CO₂ with respect to the local atmosphere (ranging from 89.9 to 206.4% with an average of 108.5 ± 3.9%; n = 30). CO₂ fluxes showed high spatial variability across the 30 clusters, ranging from -33.7 to 445.0 mmol m⁻² d⁻¹ with an average value of 170.4 ± 21.8 mmol m⁻² d⁻¹ (n = 30; Supplementary Fig. 9).’ (Page 2, lines 6-14 in the Supplementary materials).

Fig. R2 Seasonal dynamics of CO₂ concentrations from thermokarst lakes on the Tibetan Plateau. Box plots present the 25th and the 75th quartile (interquartile range), and whiskers indicate the data range among 30 clusters of thermokarst lakes sampled

in this study. The different colors represent thermokarst lakes distributed in various grassland types (alpine steppe, alpine meadow and swamp meadow).

[Comment 16] L147: Is the global lake (1.1-6.4%) referring to the contribution of CH₄ to total carbon emissions (i.e. 7.3% in this study) or CO₂ equivalent emission (i.e. 44.6% in this study)?

[Response] Sorry for the confusion. Global lake (1.1-6.4%) referred to the contribution of CH₄ to total carbon emissions (i.e. 7.3% in this study) rather than CO₂ equivalent emission (i.e. 44.6% in this study). Nevertheless, following the second reviewer's suggestion (**Comment 22: The absolute C emission is more relevant than the ratio**), we have downplayed the argumentation about the flux proportion. **Hence, this paragraph has been reorganized and this sentence has deleted in the revised MS.** Thanks for your understanding!

[Comment 17] L183-184: There seems to be a word missing from the first part of this sentence.

L193: delete consumption (CH₄ oxidation is the same thing)

[Response] Sorry for the carelessness. As mentioned above, we have downplayed the argumentation about the flux proportion and thus deleted this sentence in the revised MS. Thanks for your understanding!

[Comment 18] L210-212: This sentence is confusing and could be simplified. E.g. decrease in O₂ may increase CH₄ production and keep CH₄ produced from being oxidized and thus increase the CH₄/CO₂ ratio. CH₄ release is mentioned here and I am wondering if the authors have explored how low atmospheric pressure impacts CH₄ ebullition, this could be an interesting point to mention here.

[Response] Sorry for the confusion. As mentioned above, we have downplayed the argumentation about the flux proportion, and focused on absolute carbon emission. Thus, **we have reorganized the paragraph, and deleted this sentence in the revised MS** Thanks for your understanding!

In regard to **the question how atmospheric pressure impacts CH₄ ebullition, we have added the related discussion in the revised MS** as follows: 'The high

contribution of ebullition to the total CH₄ flux might be potentially explained by low atmospheric pressure on the Tibetan Plateau. Due to the high elevation, atmospheric pressure across the study area has a mean value of 60.9 kPa: roughly three-fifths of that at sea level (Supplementary Table 1). Two ways in which this low atmospheric pressure could result in a larger contribution of ebullition to the total CH₄ flux. On the one hand, according to the Henry's law, the lower atmospheric pressure causes lower CH₄ solubility in the water (Wiesenburg and Guinasso 1979), which can be unfavorable for CH₄ diffusion in the water column and thus impels CH₄ to be transported from the lakebed to the atmosphere in the form of bubbles. On the other hand, atmospheric pressure can directly affect bubble formation. Bubbles containing CH₄ occupy vertical tubes within the lake sediments (Martens and Klump 1980). The lower air pressure will benefit the vertical expansion of these bubbles and promote their escape from the lake sediment (Mattson and Likens 1990), and is thus associated with greater ebullition of CH₄ (Johnson et al. 1990). Consistently, the proportion of ebullition to total CH₄ fluxes was negatively associated with air pressure ($R^2 = 0.42$, $p < 0.001$; Fig. R3a) but positively correlated with elevation across sampling clusters ($R^2 = 0.42$, $p < 0.001$; Fig. R3b).’ (Pages 7-8, lines 148-163).

Fig. R3 Relationships of the contribution of ebullition to total CH₄ fluxes with atmospheric pressure (a) and altitude (b). Outliers (grey points) were excluded from the regression analysis based on Boxplot Procedures. The black lines and shades represent the regression lines with 95% confidence intervals. $***p < 0.001$.

[Comment 19] L216: See comment L132, if CO₂ concentrations were measured it would be interesting to know if they compare to lakes with similar pH.

[Response] Good suggestion! Nevertheless, as mentioned above, combining the second reviewer's suggestion (**Comment 22: *The absolute C emission is more relevant than the ratio***), we have reorganized the paragraph and thus deleted this sentence in the revised MS. Thanks for our understanding!

[Comment 20] L218: Remove “Considering that” and end sentence after (Fig 5).

L230: remove these

L249: locating → located

L251: but still have the potential to contribute to CH₄ emissions

[Response] Done as suggested.

[Comment 21] L247-251: Are there estimates of thermokarst lake expansion/loss in this region? In addition to the 20% non-grassland lakes not accounted for, there are likely new lakes.

[Response] Yes, as the reviewer speculated, it has been reported that thermokarst lake area on the Tibetan Plateau exhibited an overall increase by 113.1% in recent decades (Li et al. 2022). **We have emphasized this point in the revised MS as follows:** ‘Second, uncertainties exist in the determination of the area of the thermokarst lakes used in the regional carbon budget. Particularly, newly formed thermokarst lakes and thermokarst lakes located in desert and barren land were not considered, but they do have potential to contribute to CH₄ emissions (Mu et al. 2016). Meanwhile, theomarkast lake map used in this study suffers from uncertainties because it is based on low resolution and single satellite images and without consideration of ground ice content (Wei et al. 2021). It is thus essential to incorporate multi-satellite, higher spatial resolution remote sensing data (e.g. GF-2 and Planetscope) and ground ice content to re-evaluate thermokarst lake area and its temporal dynamics in the future. Based on the updated lake distribution and the expansion rate, an improved estimation by considering thermokarst lakes distributed in other ecosystems and fresh thermokarst lakes is needed to obtain more accurate prediction of carbon release from alpine thermokarst lakes on the Tibetan Plateau.’ (Pages 13-14, lines 291-302).

[Comment 22] L256: Could consider making a back of the envelope calculation for ice-covered period. Is there any data from below ice for lakes in this region that could

be used for such an estimate? In Denfeld et al. 2018 (doi: 10.1002/lol2.10079), Figure 5, shows % CH₄ ice cover for a global set of lakes. Could consider using the average value, however, none of the lakes in that study are from this region, so perhaps not really relevant to use.

[Response] Following the reviewer's suggestion, we estimated CH₄ emissions from alpine thermokarst lakes during the ice-cover period (28.3 Gg CH₄ yr⁻¹) using the average annual contribution of CH₄ emissions during ice-cover period from high-latitude lakes (Denfeld et al. 2018). Nevertheless, as stated by this reviewer, the rough extension of the results observed from the northern lakes to thermokarst lakes on the Tibetan Plateau may lead to potential uncertainties. Thus, more measurements from high-altitude permafrost region during the ice-cover period are needed in the future. See our responses to **Comment 4** for more detailed information.

[Comment 23] L265: *First study ever or in this region?*

[Response] First study in this region. Following the reviewer's suggestion, we have rewritten this sentence as follows: *'Moreover, this is the first study ever to evaluate the landscape-level radiocarbon ages of thermokarst lake CH₄ emissions in this alpine permafrost region.'* (Page 14, lines 309-311).

[Comment 24] **Fig 1:** *What are the lines in Fig 1B?*

[Response] Sorry for this confusion! In Fig. 1B of the previous MS, the corresponding values of light green and dark green lines represent the ebullition and diffusion CH₄ fluxes across 30 clusters, respectively. We have clearly stated this point in the Figure legend of the revised MS (Page 35, lines 805-807).

[Comment 25] **Fig 1 and Fig 2:** *It seems there are 30 data points and thus the figure is showing the flux for each cluster, is this correct? Please clarify in the figure legend.*

[Response] Yes, there are 30 data points and Figs. 1-2 show CH₄ fluxes at each cluster. We have clearly stated this point in the Figure legend of the revised MS (Page 35, lines 795-796 and lines 806-807).

[Comment 26] L457: *The Tibetan Plateau, the largest alpine permafrost region in the world, has an average elevation over 4000 m above sea level.*

[Response] Done as suggested.

[Comment 27] L466: *So is the region snow/ice covered from mid Oct to mid-May? Could mention the average number of days snow/ice covered, if known.*

[Response] Based on our field survey, the break of lake ice on the Tibetan Plateau occurs in the middle of April to early May. The freeze onset and up of lake ice appears in the early November and the late of November, respectively. Thus, **the mean ice-cover duration is around 160 days (Guo et al. 2018)**. We have clearly stated this point in the revised MS (Page 15, lines 346-347).

[Comment 28] L480-488: *See comment L81 above. So in each cluster 4 lakes were selected = 120 lakes.*

[Response] Yes, four thermokarst lakes were selected at each cluster and 30 clusters were selected across a 1,100 km transect on the Tibetan Plateau. Total 120 thermokarst lakes were thus sampled in this study (**4 lakes/cluster × 30 clusters**; Fig. R1). We have clearly stated this point in the revised MS (Page 16, lines 354-356).

[Comment 29] L488-490: *“increased the probability of capturing ebullition” makes it sound like the bubbles were targeted and not randomly captured. Rather by taking spatial variability into account diffusive flux and ebullition were both accounted for, making the CH₄ emission estimates robust.*

[Response] Sorry for the inappropriate description in the original MS. In fact, the sampling of multiple locations within multiple lakes in each cluster is to **take spatial variability into account for diffusive flux and ebullition, rather than targeting at capturing the bubbles** (Fig. R1). To avoid the confusion, **we have revised this sentence as follows**: ‘*This sampling at multiple locations within multiple lakes allowed us to consider the spatial variability of carbon fluxes in each cluster.*’ (Page 16, lines 359-361).

[Comment 30] L492-500: *so there were 130 readings (150 s – 20 s equilibration, 1 per s)? What is the sensitivity of the analyzer, i.e. 1 sec readings seem rather quick. But it seems flux is calc from two sampling times, i.e. 20 and 170 s, so over a 2.5 min period. This could be clarified here. Why was this amount of time chosen?*

[Response] Sorry for the unclear description. In this study, flux measurements were made for 170 sec at each sampling location. Before measurements, an equilibration period (20 sec) was adopted to eliminate the disturbance of surface boundary layer induced by chamber deployment. **Thus, CH₄ and CO₂ concentration were actually recorded 150 sec at 1 sec interval (i.e., 150 readings).** The concentrations of CH₄ and CO₂ were determined by micro-portable gas analyzers (GLA131-GGA, ABB Inc., Saint-Laurent, CA) which is of high **precision with < 0.9 ppb CH₄ (1 sec) and 0.35 ppm CO₂ (1 sec), and measurement rates ranged from 0.01 to 10Hz.** (<https://new.abb.com/products/measurement-products/analytical/laser-gas-analyzers/laser-analyzers/lgr-icos-portable-analyzers/lgr-icos-microportable-analyzers-gla131-series/lgr-icos-portable-analyzers-gla131-gga>).

During the measurements of CH₄ and CO₂ fluxes, **1 sec interval was chosen to explore the change of gas concentration over time within the chamber in real time. 20 sec equilibration period was used on the basis of the fact that the CH₄/CO₂ concentration within the chamber can rise or decrease steadily. Measurement time (170 sec) was adopted to avoid CO₂ accumulation within the floating chamber and subsequent inhibition on CO₂ emission.** We have rephrased the related descriptions to clarify our sampling method, and provided detailed information concerning the precision and measurement rates of gas analyzers and the reason why these times were chosen in the revised MS (Pages 16-17, lines 368-380).

[Comment 31] L505: CH₄ concentrations *in the water*.

[Response] Done as suggested.

[Comment 32] L507: What was the ratio of air-water for the headspace extraction? Was the headspace extracted in the field? If using the same exact method as reference 56 state that so that the reader knows they can look into the details there.

[Response] Dissolved CH₄ and CO₂ concentrations in the water were determined based on the improved procedure provided by Bastviken et al. (2010). In particular, in this study, **1:1 ratio of air-water was adopted for the headspace extraction, and the headspace extraction was conducted in the field.** Following the reviewer's comment, we have added the detailed descriptions in the revised MS as follows: 'Specifically,

during flux measurements, 50 ml of surface water from a depth of 0–10 cm was collected with a 100 ml syringe (1:1 ratio of air-water). Subsequently, 50 ml of pure N₂ was injected into the syringe to create 50 ml of headspace. The syringe was immediately shaken for 5 min to equilibrate the headspace in the field. The headspace sample was then injected into a vacuumed airtight vial and transported to the laboratory for analysis of CH₄ and CO₂ concentrations using a gas chromatograph (Agilent 7890A, Agilent Technologies Inc., Santa Clara, Canada). Dissolved CH₄ and CO₂ concentrations were calculated using Henry's Law adjusted for temperature and atmospheric pressure (Bastviken et al. 2010).' (Pages 17-18, lines 389-397).

[Comment 33] L507-509: See L132 above. It seems CO₂ concentration was measured.

[Response] Yes, as mentioned above, CO₂ concentration was indeed measured in this study. Combining suggestions from this reviewer (Comment 12: L132: Was CO₂ concentration measured or only flux? If measured could be nice to include.) and the second reviewer (Comment 22: The absolute C emission is more relevant than the ratio.), we have added the description for measurements of CO₂ concentrations, saturations and fluxes in the revised MS (Pages 16-18, lines 363-401).

[Comment 34] L522: How was wind speed measured to determine the Sc?

[Response] During flux measurement, we quantified wind speed, atmospheric temperature with a portable anemometer (Testo 480, Testo SE & Co. KGaA, Lenzkirch, Germany) at each cluster of thermokarst lakes. We have provided this information in the revised MS (Page 19, lines 425-427).

[Comment 35] L528: Was each 120 lakes sampled once or multiple times over the open water season? Please clarify.

[Response] Sorry for the unclear description. Each thermokarst lake was sampled five times during the ice-free period from mid-May to mid-October of 2021 (once a month). We have clarified this point in the revised MS (Page 16, lines 361-363).

[Comment 36] L567-568: Could consider putting this info at the start of the section (L536). How was the one lake per cluster selected, i.e. random or a certain size class?

[Response] Following the reviewer's comment, we have moved the related information

to the start of the section and clearly stated that **the lake was randomly selected within each cluster** when determining the carbon isotopic compositions of CH₄ and CO₂. in the revised MS (Page 19, lines 438-440).

[Comment 37] L574: we collected lake sediment samples

[Response] Done as suggested.

[Comment 38] L609: Could the same calculation be made for CO₂?

[Response] Yes, CO₂ fluxes could be upscaled using the same approach as CH₄. **We have performed the relevant analysis and added the corresponding methods in the revised MS as follows:** *‘We upscaled CH₄ and CO₂ fluxes from lake levels to the regional scale using Monte Carlo analysis that ran 1,000 iterations for each of the grassland types (including alpine steppe, alpine meadow and swamp meadow) where thermokarst lakes are mainly located. Each iteration randomly resampled a CH₄ or CO₂ flux (for one of the three grassland types) based on a normal distribution surrounding the mean and standard deviation. Then, to generate the total thermokarst lake CH₄ or CO₂ flux per unit of time for each grassland type, we multiplied the randomly resampled CH₄ or CO₂ flux values by the area of thermokarst lakes for each grassland type which was determined by the distribution of thermokarst lakes on the Tibetan Plateau (Wei et al. 2021). and the vegetation map derived from China’s Vegetation Atlas (Editorial Committee for Vegetation Map of China, 2001). Subsequently, we multiplied the total CH₄ or CO₂ flux per unit of time by the ice-free season duration (~200 d) to obtain the CH₄ or CO₂ emissions from each grassland type. Finally, we summed CH₄ or CO₂ emissions from each grassland type to estimate total CH₄ or CO₂ emissions.’ (Pages 23-24, lines 536-548).*

We have also added more descriptions about the results of CO₂ fluxes in the Supplementary materials of the revised MS as follows: *‘To obtain the regional estimation, we upscaled CO₂ fluxes from the lake level to the regional scale using the same approach as CH₄. The upscaling analyses demonstrated that CO₂ emissions from*

alpine thermokarst lakes on the Tibetan Plateau were 2084.7 Gg (10^9 g) $\text{CO}_2 \text{ yr}^{-1}$ (Table 1).’ (Page 2, lines 19-23 in Supplementary materials).

[Comment 39] Extended Data Figure 1: The total number for each grassland type is reported in the caption, summing to 120, but it is unclear if this is the total for all lakes or for each sampling date (i.e. does each bar in the box plot contain 120 points)?

[Response] Sorry for the confusion. In the figure of previous MS, five bars indicate 5 times’ sampling, and 120 points in each bar represent CH_4 concentrations from 120 thermokarst lakes. To keep consistent with other figures in this study, we have redrawn this figure which showed seasonal patterns of the CH_4 concentrations from thermokarst lakes at 30 clusters instead of 120 thermokarst lakes in the revised MS ($n = 30$). That is, the points in each bar of the redrawn figure represent the mean CH_4 concentrations of four lakes at each cluster (Fig. R4). We have also clearly stated this point in the figure legend of the revised MS (Page 8, lines 62-64 in the Supplementary materials).

Fig. R4 Seasonal dynamics of CH_4 concentrations from thermokarst lakes on the Tibetan Plateau. Box plots present the 25th and the 75th quartile (interquartile range), and whiskers indicate the data range among 30 clusters of thermokarst lakes sampled in this study. The different colors represent thermokarst lakes distributed in various grassland types (alpine steppe, alpine meadow and swamp meadow).

[Comment 40] Extended Data Figure 2: If I understand correctly each bar is n=30, so does that mean the average CH₄ and CO₂ flux from each cluster is reported for each data point in this graph?

[Response] Yes, mean CH₄ and CO₂ fluxes across 30 clusters are shown among different sampling period in this Figure. To avoid this confusion, **we have clearly stated this point in the figure legend of the revised MS as follows:** ‘Data show the average CH₄ and CO₂ fluxes across 30 clusters of thermokarst lakes in different sampling periods. Error bars represent standard error (n = 30).’ (Page 9, lines 70-72 in the Supplementary materials).

[Comment 41] Extended Data Table 1: The area ∅ Lake area in table heading

[Response] Done as suggested.

Thanks again for this reviewer’s insightful and professional review. These comments inspired us to have a deeper thinking on both field sampling and results integration, and thus guided us to conduct a thorough revision of the original MS. To address these insightful comments, **we described the field sampling strategies in more detail, added the additional information of CO₂ concentrations, saturations and fluxes, and estimated CH₄ emission during the ice-cover period.** By doing so, we feel that the revised MS has been greatly improved and expect that the reviewer will be satisfied with the revised manuscript. Thank you!

Responses to Reviewer #2

[Comment 1] This paper investigates the CH₄ (and partly CO₂) emission from thermokarst lakes located at the Tibet plateau. The region holds a large part of the alpine permafrost globally, and experience changes in permafrost dynamics that could potentially release substantial amounts of carbon to the atmosphere, directly from land or from downstream aquatic systems. The carbon emission from thermokarst lakes of the region is poorly constrained. Thus, considering that my concerns can be adequately addressed, the study is novel and important for the larger scientific community focusing on carbon cycling and climate change.

[Response] Many thanks for the reviewer's positive and insightful comments. These comments listed below help us to have a deeper thinking on this issue, and thus guided us to conduct a thorough revision of the original manuscript. We really appreciate this professional review which greatly improved our paper. Thank you! Detailed modifications please see our responses to the following comments.

Major comments:

[Comment 2] My main comment is the large focus on the CH₄:CO₂ balance and comparing CH₄ emission (not CO₂) with the CH₄ exchange of terrestrial systems. In order to understand the role of the lakes in the overall C balance of the region you need to focus on the absolute C emission, and compare this with the net C exchange of the terrestrial systems.

[Response] Very good comment! We agree with the reviewer that more focus should be on absolute carbon emissions, and comparison with the net carbon exchange of the terrestrial ecosystems. Following from this reviewer's comment, we have downplayed the argument about the relative proportion of CH₄ vs. CO₂ flux, and focused on absolute carbon emission in the revised MS. Based on this point, we have reorganized the whole manuscript to concentrate on absolute carbon emissions, especially CH₄. Specifically, we focused on the multiple characteristics on CH₄ emission, release pathway, radiocarbon ages, production pathways and methanogens. Meanwhile, we added more detailed information about CO₂ emission. In additions, we also compared the carbon emission of thermokarst lakes with net carbon exchange of alpine grasslands on the Tibetan Plateau, illustrating that carbon emissions from thermokarst lakes could offset 11.9% of net carbon exchange (*i.e.*,

carbon sink) for alpine grasslands on the Tibetan Plateau (Pages 5-13, lines 94-276). Taken together, we feel that the theme of this manuscript is more significant after revision. Many thanks for this reviewer's insightful suggestion!

[Comment 3] I also have a major comment regarding the collection of data. The sampling program is impressive and ensure that the data collected is fairly representative for the carbon emission from thermokarst lakes of the region. They sampled 120 lakes of different size distributed along an extensive (1100km) transect. Sampling was carried out monthly (5 times) during the open water season using established methods. However, additional details (see below) is needed to ensure that the sampling did not introduce and bias in the results.

[Response] Thanks for this reviewer's recognition and reminder. Following the reviewer's comment, **we provided more details about our sampling approach.** Specifically, to address the comments [23-25] regarding data collection, we have added detailed descriptions in the revised MS. In particular, in regard to the comment 23 (*How were the chambers sampled to avoid disturbance*), **we deployed the chamber to sampling location with a rope to avoid the need to enter the lake and potentially disturb sediment and gas release** (Fig. R5; Pages 16-17, lines 370-372). **An equilibration period (20 sec) was then adopted to eliminate the disturbance of surface boundary layer induced by chamber deployment** (Pages 16-17, lines 377-379). In regard to the comment 24 (*How were CH₄ collected?*), **we measured in-situ CH₄ fluxes using a near-infrared laser CH₄/CO₂ analyzer, rather than manually from a boat, which can avoid fluctuating and creating turbulence** (Fig. R5; Page 16, lines 363-368). In regard to the comment 25 (*Did you measure atm CO₂ to be used in calculations?*), **atmospheric CH₄ and CO₂ concentrations were recorded with the CH₄/CO₂ analyzer when the CH₄ and CO₂ concentrations within the chamber keep steady through being flushed with ambient air** (Page 19, lines 431-433). Based on the above descriptions, the reviewer can see that our sampling did not introduce biases in flux measurements. We have clearly stated these points in the revised MS (Pages 16-17, lines 370-372; Pages 16-17, lines 377-379; Page 16, lines 363-368 and Page 19, lines 431-433).

Fig. R5 Photograph of *in-situ* flux measurement using a near-infrared laser CH₄/CO₂ analyzer. Photograph by Guibiao Yang.

Minor comments:

[Comment 4] L74-76. In the end it is not the flux proportion, but rather the absolute amount released to atm that matter. Focusing on the proportion can give wrong impression since systems with very low total emission can have high CH₄:CO₂ ratios (if CO₂ emissions are low).

[Response] Following this reviewer's comment, we have changed the main statement from the flux proportion to absolute amount released to the atmosphere, and rewritten this sentence in the revised MS as follows: *'Overall, multiple parameters obtained in this study provide benchmarks for better predicting the strength of the feedback between permafrost carbon and climate.'* (Page 2, lines 32-34). Thanks for your understanding!

[Comment 5] L78-79. I suggest to skip hypothesis 2. It is included in hypothesis 1, and hypothesis 1 is stronger since it focus on total emission (not relative).

[Response] Following the reviewer's suggestion, we have focused on the hypothesis 1 (absolute CH₄ emission), and deleted this hypothesis 2 in the revised MS.

[Comment 6] L89-92. Good to give the gas concentrations but I suggest to not continue and make interpretations of the concentration patterns. It is the gas fluxes that are important. If you still choose to discuss the concentration it would be reasonable to also discuss gas exchange velocity (k) in order to understand their individual and combined effect on gas fluxes.

[Response] Following the reviewers' suggestion, **we have simplified the description of the CH₄ concentration** to avoid making interpretations of the concentration patterns in the revised MS as follows: 'Across the 30 sampled clusters, CH₄ concentrations ranged from 107.1 to 159.4 nmol L⁻¹ with a mean of 136 ± 30 nmol L⁻¹ (n = 30; Supplementary Fig. 2a).' (Page 5, lines 94-96).

[Comment 7] L114- The discussion and conclusion based on the "2 aspects" are not fully clear. If Tibetan soils have generally younger surface soils, but deeper active layers, compared to arctic soils, would not these 2 factors work in different directions and result in less difference in age of released C?

[Response] Very good comment! We agree with this reviewer's view that two aspects (younger surface permafrost soils and deeper active layer) did work in different directions in affecting the age of released CH₄. Following comments from this reviewer and the third reviewer (**Comment 4: The "Arctic". The authors present an oversimplistic view, comparing the Tibetan plateau with the "Arctic", as if the Arctic was a uniform entity.**), we have deleted the comparison between the Tibetan Plateau and arctic region, and **focused on the potential explanations for the relatively low contribution from old carbon observed in this study from the aspect of the young permafrost carbon rather than the deep active layer in this study region.** We have stated this point in the revised MS as follows: 'Second, young permafrost carbon in this study region could also account for the CH₄ radiocarbon age. It has been reported that permafrost on the Tibetan Plateau forms relatively recently compared to other permafrost regions (Jin et al. 2020, Wang et al. 2022), and this may mean that the frozen carbon is also relatively young. This deduction is supported by the measured average radiocarbon age of surface permafrost below the active layer at 24 sites across the Tibetan Plateau (6,100 ± 880 yrs BP; n = 24; Supplementary Fig. 7; see Supplementary Note 2 for details of radiocarbon age measurements). Therefore, the relatively low

contribution of old carbon to CH₄ fluxes in alpine thermokarst lakes could also be attributed to the young permafrost carbon in this study region.’ (Page 9, lines 185-194).

[Comment 8] L117. Not obvious how deep active layer would result in less disturbance of deep soils

[Response] Sorry for this confusion. As mentioned above, compared with young surface permafrost carbon, deep active layer could work in different directions in affecting the age of released CH₄, and thus could not be used to explain the low old carbon contribution to CH₄ production from alpine thermokarst lakes on the Tibetan Plateau. Therefore, **we have deleted the related discussion about the effect of deep active layer to CH₄ radiocarbon age in the revised MS.** Thanks for your understanding!

[Comment 9] L118-119. This part is not clear. Do you mean low contribution of potential release of C as a result of further deepening of active layer?

[Response] Sorry for this confusion. As mentioned above, deep active layer could not explain why old carbon was not the dominant source for CH₄ production in thermokarst lakes on the Tibetan Plateau. Thus, **we have deleted the related discussion about the effect of deep active layer to CH₄ radiocarbon age in the revised MS.** Thanks for your understanding!

[Comment 10] L132. Please rephrase. You did not measure CO₂ fluxes, they were calculated from concentration and k data (derived from CH₄ flux and concentration measurements).

[Response] Sorry for this misunderstanding. Actually, both CH₄ and CO₂ fluxes were simultaneously measured with CH₄/CO₂ analyzer (GLA231-GGA, ABB., Canada), rather than calculated from CH₄ and CO₂ concentration and theoretical gas transfer *k* value. The *k* value was only used to calculate diffusion CH₄ flux (Bastviken et al. 2010). To avoid this confusion, we have added the descriptions for CO₂ measurements in the revised MS (Pages 16-17, lines 354-382).

[Comment 11] L142-143. This part needs additional explanation. GPP in water could be regarded to still be reasonably captured by its general effect on pCO₂ in water (and

thus on atm fluxes). Or do you mean that there were emergent plants of which CO₂ uptake were not included in chamber estimates?

[Response] Sorry for this confusion. In the original MS, we wanted to illustrate that CO₂ fluxes were measured in opaque chamber, which did not include plant CO₂ uptake (*i.e.*, *there were emergent plants of which CO₂ uptake were not included in chamber estimates*). Anyhow, in our case, **there are slight plant CO₂ uptake due to the sparse plants in thermokarst lakes on the Tibetan Plateau.** We have clearly stated this point in the revised MS as follows: ‘*Overall, despite emergent plants and their associated CO₂ uptake, alpine thermokarst lakes are still expected to a significant carbon source due to the sparsity of plants in most lakes (Supplementary Table 1).*’ (Page 2, lines 23-25 in the Supplementary materials).

[Comment 12] L146-147. *The difference between your lakes (7.3%) and global lakes (1.1-6.4%) are rather small. Given uncertainties in data I suggest to not make a major story of this difference. A tendency for high relative emission of CH₄ vs CO₂ can be noted, but should not be the main focus of the paper. As mentioned earlier the focus should be on absolute C emissions.*

[Response] Very good comment. We agree with the reviewer that the difference between your lakes (7.3%) and global lakes (1.1-6.4%) is rather small and the focus should be on absolute carbon emissions. Following the reviewer’s comment, **we have downplayed the argument about the relative proportion of CH₄ vs. CO₂ flux, and focused on absolute carbon emission in the revised MS.** Specifically, we focused on the multiple parameters on CH₄ emission, including CH₄ release pathway, radiocarbon ages, production pathways and methanogens. We also added the information about CO₂ emissions (Page 2, lines 5-25 in the Supplementary materials).

[Comment 13] L149-216. *This part is very long and focused on factors potentiality causing high fractional CH₄. Yet to understand the high ratio you also need to discuss factors that could result in relatively low CO₂.*

[Response] Very insightful comment! We agree with the reviewer’s opinion that the factors resulting in relatively low CO₂ need to be discussed to understand high fractional CH₄ to total carbon emission. Nevertheless, as mentioned above, we have downplayed the argumentation about the relative proportion of CH₄ vs. CO₂ flux, and

focused on absolute carbon emission. Based on this point, **we have reorganized this part and discussed:** what are the potential driving factors for the high CH₄ flux and the ebullition-diffusion flux ratio, the reason why modern carbon emission was observed in this region and why CH₄ flux was driven by the CO₂ reduction process, and the role of methanogens in regulating CH₄ emission in the revised MS (Pages 5-12, lines 94-254).

[Comment 14] L163. As written it is not clear if this is different compared to other lakes, e.g. to global and arctic permafrost lakes? This needs to be shown in order to propose this as an explanation for high relative proportion of CH₄ to C emission.

[Response] Very good comment! We agree with the reviewer's view that CH₄ production pathway in global lakes and arctic thermokarst lakes needs to be shown to propose it is a potential explanation for the relative high proportion of CH₄ to total carbon emission. Although many studies reported that CH₄ emissions from natural ecosystems were dominated by acetate fermentation (Bousquet et al. 2006; Kai et al. 2011), a few studies still showed CH₄ fluxes were driven by the hydrogenotrophic methanogenesis in some natural ecosystems (including arctic thermokarst lakes; McCalley et al. 2014; Walter Anthony et al. 2008). Combining suggestions from this reviewer (**Comment 22: *The absolute C emission is more relevant than the ratio***), **we have downplayed the argument about the flux proportion, and given up the discussion that CH₄ production pathway was one of drivers for the high CH₄/CO₂ ratio** from alpine thermokarst lakes.

[Comment 15] L218-223. Long and complicated sentence. Please revise.

[Response] Following the reviewer's comment, we have broken it up into two short and brief sentences in the revised MS as follows: '*Most of the lakes (~80%) are located in alpine grasslands which can be subdivided into alpine steppe, alpine meadow and swamp meadow (Wei et al. 2021)*' (Page 4, lines 59-61), and '*To upscale our lake-level measurements to regional efflux estimates, we conducted a Monte Carlo analysis to randomly sample thermokarst lake CH₄ flux for each grassland type from a normal distribution around the mean.*' (Page 12, lines 256-258).

[Comment 16] L229. It is not clear if you mean 1.4-8.4 fold greater than the total or

the relative contribution.

[Response] Sorry for the unclear description. In the original MS, we wanted to express that **the relative contribution** of CH₄ to total carbon emission was 1.4-8.4 fold greater than that from global lakes. Nevertheless, we have deleted this sentence since the revised MS tends to focus on absolute carbon emission.

[Comment 17] L237. “Expressed differently.” is maybe not the best wording since this sentence compare with other ecosystem types. Replace with "Further,.."?

[Response] Done as suggested.

[Comment 18] L239-241. This statement is not supported by the comparison in 2 previous sentences, e.g. that lake CH₄ emissions offset 15% of CH₄ sink of steppe and meadows (that cover 95% of grasslands), and roughly equal emissions from swamps (which cover only 5% of grasslands).

Further, do make this landscape scale comparison relevant and allow assessment of overall importance of the lakes for the larger C balance you should also include CO₂ exchange (of land and water) in these comparisons.

[Response] We agree with the reviewer’s view that the statement (*This demonstrates that CH₄ emissions from thermokarst lakes are an important and non-neglectable source when estimating CH₄ budget for the Tibetan Plateau.*) is not supported by the previous sentences. To address this comment, **we have deleted this statement in the revised MS, and also added the comparison between thermokarst lake carbon emission (including both CH₄ and CO₂ fluxes) and net carbon sink for alpine grasslands on the Tibetan Plateau** as follows: ‘*CH₄ emissions from thermokarst lakes are often ignored when evaluating the regional CH₄ balance across Tibetan alpine grasslands (Wei et al. 2021). However, our results illustrated that CH₄ emissions from thermokarst lakes could offset 15.3% of the CH₄ uptake from alpine steppe and meadow which cover ~95% of alpine grasslands on the Tibetan Plateau (Wei et al. 2015a, Wei et al. 2015b). Further, thermokarst lake CH₄ emissions were equivalent to ~50–150% of the CH₄ emissions from swamp meadow which occupies ~5% of Tibetan alpine grasslands (Wei et al. 2015a, Ding et al. 2016). The incorporation of CO₂ fluxes gives an estimate of the overall carbon emissions (CH₄ + CO₂) in thermokarst lakes of 4.2 Tg (10¹² g) CO₂-e yr⁻¹ (Table 1), which could offset 11.9% of net carbon sink in*

Tibetan alpine grasslands (37.1 Tg CO₂-e yr⁻¹) (Yan et al. 2015). Taken together, these results demonstrate that any assessment of the carbon budget in this climate-sensitive region is incomplete without considering the significant carbon emissions, particularly CH₄ source from thermokarst lakes.’. (Pages 12-13, lines 265-276).

[Comment 19] L251. Provide reference to support this statement about that these lakes are contributors to emissions.

[Response] The related reference (Mu et al. 2016 Journal of Limnology; doi: 10.4081/jlimnol.2016.1346) has been provided to support this statement in the revised MS (Page 13, line 294).

[Comment 20] L253-255. Are the lakes deep enough to allow significant gas accumulation under ice? What are the mean depth of these lakes and the thickness of ice in winter? Fig 1 suggest that the lakes are very shallow and would not allow significant gas accumulation in winter. Also, you have one reference for statement of particularly high CH₄ release following ice-out. Good to refine statement and include other references that do not show clear support of this (see compilation by Denfeld et al. 2018 L&Oletters, doi: 10.1002/lol2.10079). Also, the lakes studied by Walter et al are likely quite different compared to your lakes (depth, permafrost conditions...).

[Response] We agree with the reviewer that gas accumulation from thermokarst lakes on the Tibetan Plateau during winter may not necessarily have a high contribution to annual carbon emission. Particularly, thermokarst lakes on the Tibetan Plateau are characterized by relatively small and shallow compared with those studied by Walter Anthony et al. The average depth of thermokarst lakes selected in this study was only ~0.6 m. Given that CH₄ emissions is related to lake depth in winter (Ducharme-Riel et al. 2015), the shallowness of thermokarst lakes on the Tibetan Plateau may not lead to a considerable percentage of CH₄ accumulation during winter. Based on this point, we have deleted this statement in the revised MS.

[Comment 21] L256. Based on past comment I am not convinced this is true, and I suggest to not include it. Including ice-off period could however increase absolute C emission by both CO₂ and CH₄, so this part can be revised accordingly to reflect this.

[Response] As mentioned above, we agree with the reviewer that CH₄ is less likely to

be considerably accumulated during winter across our study area, but annual carbon emissions can still be increased if the ice-melt carbon emissions are included. Combining suggestions from this reviewer and the first reviewer (**Comment 22:** *Could consider making a back of the envelope calculation for ice-covered period. In Denfeld et al. 2018 (doi: 10.1002/lo2.10079), shows % CH₄ ice cover for a global set of lakes. Could consider using the average value.*), **we have made a rough evaluation about ice-cover carbon emission, and clearly stated this point in the revised MS as follows:** ‘To make a rough evaluation of ice-cover carbon emission, we applied the average percent contribution of ice-cover to annual carbon emissions from high-latitude lakes (CO₂: 17%; CH₄: 27%) (Denfeld et al. 2018). These increased annual emissions estimate from alpine thermokarst lakes on the Tibetan Plateau to 427.0 Gg CO₂ yr⁻¹ and 28.3 Gg CH₄ yr⁻¹, respectively. Nevertheless, given the potential differences between high-latitude and high-altitude permafrost regions, such as climate, thermokarst lake depth and ice duration (Wang et al. 2022), the annual contribution of carbon emissions during the ice-cover period observed from high-latitude lakes may not be simply applied to thermokarst lakes in high-altitude permafrost region. More measurements during the ice-cover period are thus needed to further advance our understanding of CH₄ emissions in alpine thermokarst lakes.’ (Page 13, lines 279-290).

[Comment 22] L274-275. *The absolute C emission is more relevant than the ratio. There are lakes with CH₄ release but CO₂ uptake which with this reasoning would suggest that they are very important despite they may be negligible in terms of overall C emissions. I would downplay this argumentation.*

[Response] We agree with the reviewer that the absolute carbon emission is more relevant than the ratio. Following the reviewer’s suggestion, **we have downplayed the argument about the relative proportion of CH₄ vs. CO₂ flux, focused on absolute carbon emission, and deleted this sentence in the revised MS.**

[Comment 23] L476-. *Give more details to ensure nonbiased data. How were the chambers sampled to avoid disturbance, which is easily created in small shallow systems (where turbulence enhance fluxes).*

[Response] The disturbance of chamber deployment was reduced by the following two aspects. **On the one hand, we deployed the chamber to sampling location with a**

rope to avoid the need to enter the lake and potentially disturb sediment and gas release. (Fig. R5; Bouchard et al., 2015) (Pages 16-17, lines 370-372). On the other hand, an equilibration period (20 sec) was adopted to eliminate the disturbance of surface boundary layer induced by chamber deployment (Peng et al., 2017) (Page 17, lines 377-379). We have added this information in the revised MS.

[Comment 24] L492-493. How were CH₄ collected? In situ sensor or manually from boat? The later can create turbulence and bias flux estimates so important that you clarify how it was done.

[Response] Actually, we did not collect gas samples and *in-situ* CH₄ fluxes were determined using a near-infrared laser CH₄/CO₂ analyzer (GLA231-GGA, ABB., Canada), rather than manually from a boat in this study (Fig. R5). We have clearly stated this point in the revised MS (Page 16, lines 361-368).

[Comment 25] L525. Did you measure atm CO₂ to be used in calculations? Please provide details.

[Response] Yes, we did. Both atmospheric CH₄ and CO₂ concentrations were recorded with the CH₄/CO₂ analyzer (GLA231-GGA, ABB., Canada; Fig. R5) as the steady values obtained when being flushed through with ambient air. We have clearly stated this point in the revised MS (Page 19, lines 431-433).

[Comment 26] L610. Did you not also upscale CO₂ fluxes?

[Response] Yes, we upscaled CO₂ fluxes using the same approach as CH₄. We have added the description for CO₂ spatial upscaling and its uncertainty analyses in the revised MS as follows: *‘We upscaled CH₄ and CO₂ fluxes from lake levels to the regional scale using Monte Carlo analysis that ran 1,000 iterations for each of the grassland types (including alpine steppe, alpine meadow and swamp meadow) where thermokarst lakes are mainly located. Each iteration randomly resampled a CH₄ or CO₂ flux (for one of the three grassland types) based on a normal distribution surrounding the mean and standard deviation. Then, to generate the total thermokarst lake CH₄ or CO₂ flux per unit of time for each grassland type, we multiplied the randomly resampled CH₄ or CO₂ flux values by the area of thermokarst lakes for each grassland type which was determined by the distribution of thermokarst lakes on the*

Tibetan Plateau (Wei et al. 2021) and the vegetation map derived from China's Vegetation Atlas (Editorial Committee for Vegetation Map of China, 2001). Subsequently, we multiplied the total CH₄ or CO₂ flux per unit of time by the ice-free season duration (~200 d) to obtain the CH₄ or CO₂ emissions from each grassland type. Finally, we summed CH₄ or CO₂ emissions from each grassland type to estimate total CH₄ or CO₂ emissions.' (Pages 23-24, lines 536-548).

We have also added more descriptions about the results of regional CO₂ emission in the Supplementary materials of the revised MS as follows: *'To estimate the magnitude of the total CO₂ emissions from thermokarst lakes across our study area, we upscaled CO₂ fluxes from the lake level to the regional scale using the same approach as CH₄. The upscaling analyses demonstrated that CO₂ emissions from alpine thermokarst lakes on the Tibetan Plateau were 2084.7 Gg (10⁹ g) CO₂ yr⁻¹ (Table 1).'* (Page 2, lines 19-23 in Supplementary materials).

Taken together, we are very grateful to the reviewer for the insightful comments on our manuscript! **As mentioned above, we have downplayed the argumentation about the relative proportion of CH₄ vs. CO₂ flux, and focused on absolute CH₄ emission. Additionally, we have provided detailed information about flux measurements.** By addressing these comments, we feel that the revised MS has been greatly improved. Thank you!

Responses to Reviewer #3

Major comments:

[Comment 1] The manuscript “Vital contribution of methane to total carbon emissions from thermokarst lakes” by Yang et al., presents measurement of CH₄ and CO₂ fluxes from lakes located on the Tibetan plateau. These data are highly valuable and will be of interest to the community. However, the manuscript suffers from lots of imprecisions, both in the data interpretation and reference to the literature. There are a few examples (listed below) where the interpretation given by the authors does not match the presented results. The authors also generally oversimplify the literature and compare their study with “the arctic”, when there is obviously no unique behavior of high latitude lakes, but a wide diversity of geologic and climatic contexts.

[Response] Many thanks for the reviewer’s comments on our MS. These comments, together with those listed below inspired us to have a deeper thinking on this issue, and thus guided us to conduct a thorough revision of the original MS. To address the reviewer’s comments listed below, we made the following two major changes:

1. We strengthened the relevant interpretation about why old carbon was not the dominant source for CH₄ production **in most thermokarst lakes** on the Tibetan Plateau (*Comment 3*) and how atmospheric pressure impacts CH₄ ebullition (*Comment 13*), and carefully addressed issues about the reference throughout the manuscript (*Comments 11-12*).
2. We deleted the comparison of thermokarst lakes between the Tibetan Plateau and arctic region (*Comment 4*). Instead, the revised MS focused on discussing the characteristics of CH₄ emissions in alpine thermokarst lakes on the Tibetan Plateau.

By doing so, we felt that the revised MS had been significantly improved. For the detailed response to each comment, please see below.

[Comment 2] Ebullition vs diffusion: A major result emphasized in this manuscript is the high contribution of ebullition fluxes for CH₄. Yet, this partitioning is based on the use of a theoretical k value. This implies a huge uncertainty in the partitioning between the two pathways of CH₄ emission. The ebullition fluxes could have been identified looking at the continuous records of [CH₄] in the chamber. The authors should include examples of fluxes that have been attributed to ebullition and diffusion.

[Response] Very good comment! Following the reviewer’s suggestion, **we provided**

two visible examples to display a mass of ebullition occurring at thermokarst lakes involved in this study (Supplementary Video and Fig. R6) in the revised MS (Page 7, lines 140-142). These examples supported this reviewer's opinion that ebullition can be identified by looking at the continuous records of CH₄ concentrations in the chamber. In regard to the uncertainty involved in the measurement of CH₄ diffusion fluxes, **we would like to mention that this method is widely used to calculate CH₄ diffusion flux from the arctic thermokarst lakes** (Serikova et al. 2019 *Nature Communications*; Walter Anthony et al. 2016 *Nature Geoscience*) **and alpine rivers** (Zhang et al. 2020 *Nature Geoscience*). Nevertheless, to further reduce this uncertainty, other estimation approaches (e.g. bubble traps) should be attempted to quantify CH₄ ebullition and diffusion fluxes in the future. We have clearly stated this point in the revised MS (Pages 18-19, lines 418-423). Thanks for your understanding!

Supplementary Video: Video of visible micro-bubbles rising from sediment to the water surface. This video was shot by Guibiao Yang at the shore of a thermokarst lake located in Qumarlêb County.

Fig. R6 Change of CH₄ concentration in the chamber at the shore of thermokarst lake located in Qumarlêb County. In the shade area, the rapid increase in CH₄ concentration is due to the emergence of bubbles.

[Comment 3] Origin of emitted CH₄. The conclusion stated by the authors: “The measured CH₄ radiocarbon ages were in a wide range between modern and 3,810 years before present (yrs BP; n = 24; Fig.2A and Supplementary Table 2), indicating that CH₄ production was mainly from recently fixed organic carbon” does not match the results. Yet, 3810 BP is not really “recently fixed organic carbon”. When the authors cite an average age of 6100 yrs BP for Tibetan soils, does it mean that those soils (frozen? we do not know) contribute to half of the emitted carbon in form of CH₄?

[Response] Sorry for the inaccurate description. As stated by the reviewer, 3810 BP is not recently fixed organic carbon, in which old carbon may contribute much to CH₄ emissions although we can’t quantify the specific proportion based on current data. Nevertheless, we would like to mention that the contribution of permafrost carbon is small in most sampling lakes. In support of this argument, the measured CH₄ radiocarbon ages only averaged 325.8 yrs BP. Of them, 46% of thermokarst lakes had modern age of CH₄ emissions and only two lakes had higher CH₄ radiocarbon ages than 1,000 yrs BP (Fig. R7). To clearly state this point, we have added more descriptions about the results of CH₄ radiocarbon ages as follows: “The results showed relatively young CH₄ radiocarbon age, ranging from -360 to 3,810 years before present (yrs BP; Fig. 4b). The mean CH₄ radiocarbon age was only 325.8 yrs BP, and 46% of thermokarst lakes had modern (defined here as created after 1950) age of CH₄

emissions (Supplementary Table 2). Only two lakes had higher CH₄ radiocarbon ages than 1,000 yrs BP, while radiocarbon ages were between modern and 1,000 yrs BP in the remaining 22 samples. These observations indicated that old carbon was not the dominant source for CH₄ production in most thermokarst lakes on the Tibetan Plateau.’ (Page 8, lines 170-175), and **reorganized this argument** as follows: “old carbon was not the dominant source for CH₄ production **in most thermokarst lakes** on the Tibetan Plateau.’ (Page 8, lines 176-177). Thanks for your understanding!

Fig. R7 Density distribution of CH₄ radiocarbon ages from the investigated thermokarst lakes on the Tibetan Plateau. The line indicates the CH₄ radiocarbon age at each sampling lake ($n = 24$).

[Comment 4] The “Arctic”. The authors present an over-simplistic view, comparing the Tibetan plateau with the “Arctic”, as if the Arctic was a uniform entity. This is especially striking when the authors cite and a unique AL depth (0.7) and an average age of 14,000+/-4000 yrs BP for emitted CH₄ from the Arctic.

[Response] Very good comment! We agree with this reviewer’s view that we present an over-simplistic view as the Arctic is a uniform entity. In fact, as stated by the reviewer, arctic permafrost region is characterized by large spatial heterogeneity in active layer thickness (Brown et al, 2000) and soil carbon ages (Dutta et al. 2006; Knoblauch et al. 2013; Gentsch et al. 2018). Based on this point, **we have deleted the comparison about thermokarst lakes between the Tibetan Plateau and arctic region, and**

reorganized this part by discussing the following points in the revised MS: what are the potential driving factors for the high CH₄ flux and the ebullition-diffusion flux ratio? why was modern carbon emission observed in most sampling lakes? why CO₂ reduction pathway dominated CH₄ production? what's the potential role of methanogens in mediating CH₄ emission? (Pages 5-12, lines 94-254). Thanks for your understanding!

Minor Comments:

[Comment 5] L34-37. This long sentence could be split.

[Response] Combining this with suggestions from the second reviewer (**Comment 22: The absolute C emission is more relevant than the ratio**), we have downplayed the argument about the flux proportion and focused on absolute carbon emission. Thus, we have deleted this sentence in the revised MS. Thanks for your understanding!

[Comment 6] L81. “120 lakes in 30 clusters”: The authors could precise what was the timing of the measurements (time in the season, time of the day), and how long did a campaign last?

[Response] Following the reviewer's suggestion, we have clearly stated what was the time of the measurements in the season and how long did a campaign last in the revised MS as follows: *'In this context, we conducted a large-scale sampling campaign across 120 thermokarst lakes in 30 clusters (four lakes in each cluster: 4 lakes/cluster × 30 clusters) along a 1,100 km transect on the Tibetan Plateau (Fig. 1a-c). Each lake was sampled five times at monthly intervals during the ice-free period from mid-May to mid-October of 2021 (Fig. 1d-h), with each campaign lasting ~25 days.'* (Page 4, lines 76-80). As done in literature (Laurion et al. 2010), carbon flux was measured between 9: 00 am – 18: 00 pm in this study. We also have mentioned this point in the *Method* section of the revised MS (Page 16, lines 361-368).

[Comment 7] L100-102. Comparison with Wik et al., 2016. In this publication the high range corresponds to the measured values in this study.

[Response] Following the reviewer's comment, we have revised this sentence as follows: *'The mean CH₄ flux during the ice-free season was $13.4 \pm 1.5 \text{ mmol m}^{-2} \text{ d}^{-1}$. This value is at the high end of the range reported from Arctic thermokarst water bodies*

regarded as hotspots for CH₄ release (Wik et al. 2016; Kuhn et al. 2021).’ (Page 6, lines 108-110).

[Comment 8] L127. presents “Seasonal and regional patterns of CH₄ concentrations from thermokarst lakes in Tibetan alpine permafrost region”. Not the age of organic carbon in soils

[Response] This is a misunderstanding caused by inappropriate format. Previous MS was organized as *Nature* format. Actually, “Seasonal and regional patterns of CH₄ concentrations from thermokarst lakes in Tibetan alpine permafrost region” is presented in the Extended Data Figure 1. Supplementary Figure 1 indeed shows “Comparisons of the ages of organic carbon in surface permafrost between the Tibetan and Arctic permafrost region.”. To avoid this confusion, we have reorganized this manuscript as the format of *Nature Communications* according to the editor's suggestion.

Nevertheless, combining this reviewer’s comment (*Comment 1: The authors also generally oversimplify the literature and compare their study with “the arctic”*) and the first reviewer’s comment (*Comment 13: the comparison between arctic lake and thermokarst lake on Tibetan Plateau is unreasonable*), **we have given up this comparison between arctic lake and thermokarst lake on Tibetan Plateau and deleted this Supplementary Figure** in the revised MS. Thanks for your understanding!

[Comment 9] L155. Is -72.5+/-1.1 the mean +/- standard deviation? On the plot, d¹³C-CH₄ range from -32 to -85, it seems that the dispersion is much larger?

[Response] In the original MS, the values were reported as mean ± standard error rather than mean ± standard deviation. In fact, the data of d¹³C-CH₄ would have a larger dispersion if expressing as standard deviation of 6.1 (ranging from -83.4‰ to -62.2‰; *n* = 29), which was **similar to that reported in arctic lakes (ranging from -79.9‰ to -53.7‰ with standard deviation of 8.1‰; *n* = 26) by Walter Anthony et al. (2008)**. We have clearly stated this point (Page 5, lines 84-85: **hereafter, values are reported as mean ± standard error (SE) unless stated otherwise**) and shown the corresponding original data in the Supplementary Table 2 in the revised MS. Thanks for your understanding!

[Comment 10] L156. Do the authors have any comments on the $d^{13}C$ -CO₂ value dispersion?

[Response] As mentioned above, the values were reported as mean \pm standard error rather than mean \pm standard deviation. The $\delta^{13}C$ -CO₂ ranged between -22.9‰ and -0.5‰ with an average of -13.4‰ and standard deviation of 6.2 ($n = 29$), which was **comparable to that observed from arctic lakes (ranging from -26.2‰ to -9.6‰ with mean \pm standard deviation: -16.5 ± 6.1 ‰; $n = 6$) by Walter Anthony et al. (2008).** Original data have been shown in the Supplementary Table 2 in the revised MS.

[Comment 11] L184-186. This statement is not supported by any reference. Please include one.

[Response] As mentioned above, following the suggestion from the second reviewer (**Comment 22: The absolute C emission is more relevant than the ratio**), we have given up the argument about the flux proportion, reorganized the paragraph, and deleted this sentence in the revised MS. In addition, we have carefully checked and cited wherever the reference was needed throughout the manuscript. Thanks for your understanding!

[Comment 12] L187. Any more recent/ adequate reference for CH₄ oxidation in high latitude lakes?

L191. Any reference for the very general statement on methanotroph abundance in the Arctic?

[Response] As mentioned above, following the suggestion from the second reviewer (**Comment 22: The absolute C emission is more relevant than the ratio**), we have reorganized the paragraph, and deleted related information about methanotroph in the revised MS. In addition, we have carefully checked and cited wherever the reference was needed throughout the manuscript. Thanks for your understanding!

[Comment 13] L202. Interesting discussion, but the authors should strengthen their demonstration of the contribution of ebullition fluxes.

[Response] As mentioned above, to strengthen the demonstration of the high contribution of ebullition to total CH₄ fluxes, we provided two visible examples (Supplementary Video and Fig. R6) **to display a mass of ebullition occurring at the thermokarst lake** in the revised MS.

[Comment 14] L208-212. Unclear statement

[Response] Sorry for this poor statement. Combining suggestions from this reviewer and the second reviewer (**Comment 22: *The absolute C emission is more relevant than the ratio***), we have given up the argument about the flux proportion, and focused on absolute carbon emission in the revised MS. Thus, these sentences have been deleted in the revised MS. Thanks for your understanding!

[Comment 15] L233. In the map published by Wei et al., All lakes in a permafrost area are considered to be thaw lakes. Do the authors have any comment on this statement? What are the implication in terms of CH₄ fluxes?

[Response] Very good comment! Based on our understanding, some lakes (such as tectonic lakes and glacial lakes and so on) on the Tibetan Plateau should not belong to thermokarst lakes, and thermokarst lake area may thus be overestimated in the map by Wei et al. (2021). Meanwhile, thermokarst lakes with sizes < 500 m² were omitted in this dataset (Wei et al. 2021), which may lead to an underestimation for thermokarst lake number and area. In addition, theomarkast lake map by Wei et al. (2021) may suffer from uncertainties due to the less consideration of ground ice content. **Based on above points, we think that thermokarst lake area on the Tibetan Plateau is still of large uncertainty, which may lead to great uncertainty in regional carbon budget.** Therefore, incorporating data of high-resolution satellite images (*e.g.* GF-2 and Planetscope imageries) and ground ice content should be helpful for accurately identifying thermokarst lakes and quantifying carbon emissions from alpine thermokarst lakes. We have clearly stated this point in the revised MS (Pages 13-14, lines 291-302). Thanks for your understanding!

Overall, we are very grateful to the reviewer for the insightful comments on our manuscript! These comments guided us to have a deeper thinking on **the data interpretation, uncertainties in flux measurements, comparison of thermokarst lakes between the Tibetan Plateau and Arctic region, and reference citations and so on.** By doing so, we feel that our revised manuscript has been greatly improved, and and expect that the reviewer will be satisfied with the revised manuscript. Thank you!

References

1. Bastviken, D. et al. Methane emissions from Pantanal, South America, during the low water season: toward more comprehensive sampling. *Environ. Sci. Technol.* **44**, 5450-5455 (2010).
2. Bousquet, P. et al. Contribution of anthropogenic and natural sources to atmospheric methane variability. *Nature* **443**, 439-443 (2006).
3. Brown, J. et al. *Circum-arctic map of permafrost and ground ice conditions*. (National Snow & Ice Data Center, 1998).
4. Brown, J., Hinkel, K. M. & Nelson, F. E. The circumpolar active layer monitoring (calm) program: Research designs and initial results¹. *Polar Geogr* **24**, 166-258 (2000).
5. Chen, H. et al. Carbon and nitrogen cycling on the Qinghai–Tibetan Plateau. *Nat. Rev. Earth Environ.* **3**, 701-716 (2022).
6. Chinese Academy of Sciences. *Vegetation Atlas of China*. (Science Press, 2001).
7. Denfeld, B. A. et al. A synthesis of carbon dioxide and methane dynamics during the ice-covered period of northern lakes. *Limnol. Oceanogr. Lett.* **3**, 117-131 (2018).
8. Ding, J. et al. The permafrost carbon inventory on the Tibetan Plateau: a new evaluation using deep sediment cores. *Glob. Chang. Biol.* **22**, 2688-2701 (2016).
9. Ducharme-Riel, V. et al. The relative contribution of winter under-ice and summer hypolimnetic CO₂ accumulation to the annual CO₂ emissions from Northern Lakes. *Ecosystems* **18**, 547-559 (2015).
10. Dutta, K., Schuur, E., Neff, J. C. & Zimov, S. A. Potential carbon release from permafrost soils of northeastern siberia. *Global Change Biol* **12**, 2336-2351 (2006).
11. Holgerson, M. A. & Raymond, P. A. Large contribution to inland water CO₂ and CH₄ emissions from very small ponds. *Nat. Geosci.* **9**, 222-226 (2016).
12. Gentsch, N. et al. Temperature response of permafrost soil carbon is attenuated by mineral protection. *Global Change Biol* **24**, 3401-3415 (2018).
13. Guo, L. et al. Uncertainty and variation of remotely sensed lake ice phenology across the Tibetan Plateau. *Remote Sens.* **10**, 1534 (2018).
14. Jin, H. et al. Quaternary Permafrost in China: Framework and Discussions. *Quaternary* **3**, 32 (2020).

15. Johnson, K. M. et al. Bottle-calibration static head space method for the determination of methane dissolved in seawater. *Anal. Chem.* **62**, 2408-2412 (1990).
16. Kai, F. M. et al. Reduced methane growth rate explained by decreased Northern Hemisphere microbial sources. *Nature* **476**, 194-197 (2011).
17. Knoblauch, C. et al. Predicting long-term carbon mineralization and trace gas production from thawing permafrost of Northeast Siberia. *Global Change Biol* **19**, 1160-1172 (2013).
18. Kuhn, M. A. et al. BAWLD-CH₄: a comprehensive dataset of methane fluxes from boreal and arctic ecosystems. *Earth Syst. Sci. Data* **13**, 5151-5189 (2021).
19. Laurion, I. et al. Variability in greenhouse gas emissions from permafrost thaw ponds. *Limnol. Oceanogr.* **55**, 115-133 (2010).
20. Li, L. et al. Thermokarst lake changes over the past 40 years in the Qinghai–Tibet Plateau, China. *Front. Environ. Sci.* **10**, 1051086 (2022).
21. Martens, C. S. & Klump, J. V. Biogeochemical cycling in an organic-rich coastal marine basin — I. Methane sediment-water exchange processes. *Geochim. Cosmochim. Ac.* **44**, 471-490 (1980).
22. Mattson, M. D. & Likens, G. E. Air pressure and methane fluxes. *Nature* **347**, 718-719 (1990).
23. McCalley, C. K. et al. Methane dynamics regulated by microbial community response to permafrost thaw. *Nature* **514**, 478-481 (2014).
24. Mu, C. et al. Dissolved organic carbon, CO₂, and CH₄ concentrations and their stable isotope ratios in thermokarst lakes on the Qinghai-Tibetan Plateau. *J. Limnol.* **75**, 313-319 (2016).
25. Peng, Y. et al. Linkages of plant stoichiometry to ecosystem production and carbon fluxes with increasing nitrogen inputs in an alpine steppe. *Global Change Biol* **23**, 5249-5259 (2017).
26. Serikova, S. et al. High carbon emissions from thermokarst lakes of Western Siberia. *Nat. Commun.* **10**, 1552 (2019).
27. Wang, X. et al. Contrasting characteristics, changes, and linkages of permafrost between the Arctic and the Third Pole. *Earth-Sci. Rev.* **230**, 104042 (2022).
28. Walter Anthony, K. M. et al. Methane emissions proportional to permafrost carbon thawed in Arctic lakes since the 1950s. *Nat. Geosci.* **9**, 679-682 (2016).

29. Walter Anthony, K. M. et al. Methane production and bubble emissions from arctic lakes: Isotopic implications for source pathways and ages. *J. Geophys. Res.* **113**, G00A08 (2008).
30. Wei, D. et al. Considerable methane uptake by alpine grasslands despite the cold climate: *in situ* measurements on the central Tibetan Plateau, 2008-2013. *Glob. Chang. Biol.* **21**, 777-788 (2015a).
31. Wei, D. et al. Revisiting the role of CH₄ emissions from alpine wetlands on the Tibetan Plateau: Evidence from two *in situ* measurements at 4758 and 4320m above sea level. *J. Geophys. Res. Biogeosci.* **120**, 1741-1750 (2015b).
32. Wiesenburg, D. A. & Guinasso, N. L. Equilibrium solubilities of methane, carbon monoxide, and hydrogen in water and sea water. *J. Chem. Eng. Data* **24**, 356-360 (1979).
33. Wiesenburg, D. A. & Guinasso, N. L. Equilibrium solubilities of methane, carbon monoxide, and hydrogen in water and sea water. *J. Chem. Eng. Data* **24**, 356-360 (1979).
34. Wik, M. et al. Climate-sensitive northern lakes and ponds are critical components of methane release. *Nat. Geosci.* **9**, 99-105 (2016).
35. Wei, Z. et al. Sentinel-based inventory of thermokarst lakes and ponds across permafrost landscapes on the Qinghai-Tibet Plateau. *Earth Space Sci.* **8**, e2021EA001950 (2021).
36. Yan et al. The spatial and temporal dynamics of carbon budget in the alpine grasslands on the Qinghai-Tibetan Plateau using the Terrestrial Ecosystem Model. *J Clean Prod.* **107**, 195-201 (2015).
37. Zhang, L. et al. Significant methane ebullition from alpine permafrost rivers on the East Qinghai-Tibet Plateau. *Nat. Geosci.* **13**, 349-354 (2020).
38. Zou, D. et al. A new map of permafrost distribution on the Tibetan Plateau. *Cryosphere* **11**, 2527-2542 (2017).

Reviewer #1 (Remarks to the Author):

In Yang et al., I appreciate the effort the authors made to address my previous comments. In particular, the revisions now clarify the sampling scheme, include CO₂ data to a larger extent and put emissions into an annual context. With these revisions and as stated previously, the carbon emission reported in this study from an understudied thermokarst lake region are timely given their climate sensitivity, with results that have important implications for understanding carbon emissions from lakes globally.

Dr. Blaize Denfeld

Reviewer #2 (Remarks to the Author):

In general the authors have done a great job in addressing my comments. I just have a few rather minor comments.

L 180-184. You state that "It is well known that thermokarst lakes on the Tibetan Plateau are characterized by the small surface area, and small lakes have a high perimeter to surface area ratio, potentially increasing terrestrial loading of organic matter from surrounding plants and surface soils."

Despite small lake area and high perimeter:surface area the catchments can still be small relatively to lake area, implying relatively low terrestrial C supply per lake area. The pictures submitted suggest the landscapes sampled have dense populations of small lakes, and thus cannot have very big catchment per lake area. But, I may be wrong and you have a much better understanding of these systems. If you agree, however, I suggest that you revise this part to downplay this explanation.

L276. I would delete "particularly CH₄ source" since CO₂ was actually more important for total C emission.

Also check ms for typos and use of decimals (to detailed info at eg. L 117, 172, 263, 264).

Reviewer #3 (Remarks to the Author):

Review on revised manuscript « Methane emissions from alpine thermokarst lakes on the Tibetan Plateau: amount, origin, and methanogenic microorganisms.»

The authors have completed an extensive review. They have answered most of the concerns raised during the first review process, and substantially improved the manuscript. However, I still have concerns detailed below, that should be addressed before publication.

1. L 184 . The statement "This terrestrial organic matter are dominated by modern carbon²⁶ " is supported by the reference "Holgerson, M. A. & Raymond, P. A. Large contribution to inland water CO₂ and CH₄ emissions from very small ponds. Nat. Geosci. 9, 222-226 (2016)." This reference does not provide any data about the age of terrestrial organic matter. Please provide an adequate reference.

2. The whole discussion about carbon age is related to my main point, which was not fully addressed by the authors. Are all the studied lake thermokarst lakes? This hypothesis is uncertain, as the authors recognized in their answer. Although located in a permafrost area, some of the studied lakes might not be thermokarst lakes. Or they could have developed following thermokarst processes a long time ago. This might be part of the explanation of the modern age of emitted CH₄. This should be included when the authors discuss about the uncertainty of the regional carbon budget. This should be emphasized in the rest of the manuscript, since, there is no discussion at all on this point. The title states that data originate from thermokarst lakes only.

Responses to Reviewer #1

[Comment 1] In Yang et al., I appreciate the effort the authors made to address my previous comments. In particular, the revisions now clarify the sampling scheme, include CO₂ data to a larger extent and put emissions into an annual context. With these revisions and as stated previously, the carbon emission reported in this study from an understudied thermokarst lake region are timely given their climate sensitivity, with results that have important implications for understanding carbon emissions from lakes globally.

Dr. Blaize Denfeld

[Response] Thanks for Dr Denfeld's positive comments and recognition for our revision.

Responses to Reviewer #2

[Comment 1] In general, the authors have done a great job in addressing my comments. I just have a few rather minor comments. L 180-184. You state that "It is well known that thermokarst lakes on the Tibetan Plateau are characterized by the small surface area, and small lakes have a high perimeter to surface area ratio, potentially increasing terrestrial loading of organic matter from surrounding plants and surface soils." Despite small lake area and high perimeter:surface area, the catchments can still be small relatively to lake area, implying relatively low terrestrial C supply per lake area. The pictures submitted suggest the landscapes sampled have dense populations of small lakes, and thus cannot have very big catchment per lake area. But, I may be wrong and you have a much better understanding of these systems. If you agree, however, I suggest that you revise this part to downplay this explanation.

[Response] We agree with the reviewer that the landscapes with high dense populations of lakes may have relatively low catchment per lake area, implying relatively low terrestrial carbon supply per lake area. Nevertheless, we would like to mention that not all catchments have intensive thermokarst lakes across our study area. Anyhow, to avoid this confusion, **we followed the reviewer's suggestion to downplay this explanation in the revised MS** as follows: *'In addition, thermokarst lakes on the Tibetan Plateau are characterized by the small surface area²², and small lakes have a high perimeter to surface area ratio, which potentially increase terrestrial loading of organic matter from surrounding plants and surface soils. This*

terrestrial organic matter is dominated by modern carbon³⁴, and can thus stimulate modern carbon emissions from thermokarst lakes³⁵. (Page 9, lines 180-185).

[Comment 2] L276. I would delete "particularly CH₄ source" since CO₂ was actually more important for total C emission.

[Response] Done as suggested (Page 13, line 269).

[Comment 3] Also check ms for typos and use of decimals (to detailed info at eg. L 117, 172, 263, 264.)

[Response] We have carefully checked these issues about the typos and use of decimals throughout the manuscript.

Responses to Reviewer #3

[Comment 1] Review on revised manuscript « Methane emissions from alpine thermokarst lakes on the Tibetan Plateau: amount, origin, and methanogenic microorganisms.». The authors have completed an extensive review. They have answered most of the concerns raised during the first review process, and substantially improved the manuscript.

[Response] Thanks for the reviewer's positive comments and recognition for our revision.

[Comment 2] However, I still have concerns detailed below, that should be addressed before publication. 1. L 184. The statement 'This terrestrial organic matter is dominated by modern carbon' is supported by the reference "Holgerson, M. A. & Raymond, P. A. Large contribution to inland water CO₂ and CH₄ emissions from very small ponds. *Nat. Geosci.* 9, 222-226 (2016)." This reference does not provide any data about the age of terrestrial organic matter. Please provide an adequate reference.

[Response] Following the reviewer's comment, we have carefully consulted the relevant literature and cited it (Dean et al. 2020 *Nature Communications*; doi.org/10.1038/s41467-020-15511-6) to support this statement in the revised MS (Page 9, line 184).

[Comment 3] The whole discussion about carbon age is related to my main point, which was not fully addressed by the authors. Are all the studied lake thermokarst lakes? This hypothesis is uncertain, as the authors recognized in their answer. Although located in a permafrost area, some of the studied lakes might not be thermokarst lakes. Or they could have developed following thermokarst processes a long time ago. This might be part of the explanation of the modern age of emitted CH₄. This should be included when the authors discuss about the uncertainty of the regional carbon budget. This should be emphasized in the rest of the manuscript, since, there is no discussion at all on this point. The title states that data originate from thermokarst lakes only.

*[Response] Following the reviewer's suggestion, we have **added the explanation of the modern age of emitted CH₄ in the revised MS** as follows 'The last but not the least, some of the studied lakes may have developed following thermokarst processes a long time ago or even not be thermokarst lakes. This might also be part of the explanation of the modern age of emitted CH₄.' (Page 9, lines 185-187). We have also **added the discussion for the uncertainty of the regional carbon budget in the revised MS** as follows: 'Third, the development of thermokarst lake is not taken into account in this study. The studied lakes may cover the various thermokarst development stages or even non-thermokarst lakes, which can result in the uncertainty of the regional carbon emissions. Therefore, additional attention should be paid for the development of thermokarst lakes to further advance our understanding of CH₄ emissions in alpine thermokarst lakes on the Tibetan Plateau.' (Page 14, lines 296-301).*